# Assessment of Performance of Photocatalytic Nanostructured Materials with Varied Morphology Based on Reaction Conditions

**DOI:** 10.3390/molecules27227778

**Published:** 2022-11-11

**Authors:** Ashok Kumar Ganguli, Gajanan B. Kunde, Waseem Raza, Sandeep Kumar, Priyanka Yadav

**Affiliations:** 1Department of Chemistry, Indian Institute of Technology Delhi, Hauz Khas, New Delhi 110016, India; 2Department of Materials Science and Engineering, Indian Institute of Technology Delhi, Hauz Khas, New Delhi 110016, India

**Keywords:** photocatalyst, morphology, nanomaterials, crystal facets, environment

## Abstract

Synthesis of nanomaterials with specific morphology is an essential aspect for the optimisation of its properties and applications. The application of nanomaterials is being discussed in a wide range of areas, one of which is directly relevant to the environment through photocatalysis. To produce an effective photocatalyst for environmental applications, morphology plays an important role as it affects the surface area, interfaces, crystal facets and active sites, which ultimately affects efficiency. The method of synthesis and synthesis temperature can be the basic considerations for the evaluation of a particular nanomaterial. In this study, we have considered the aspects of morphology with a basic understanding and analyzed them in terms of nanomaterial efficacy in photocatalysis. Different morphologies of specific nanomaterials such as titanium dioxide, zinc oxide, silver phosphate, cadmium sulphide and zinc titanate have been discussed to come to reasonable conclusions. Morphologies such as nanorods, nanoflower, nanospindles, nanosheets, nanospheres and nanoparticles were compared within and outside the domain of given nanomaterials. The different synthesis strategies adopted for a specific morphology have been compared with the photocatalytic performance. It has been observed that nanomaterials with similar band gaps show different performances, which can be linked with the reaction conditions and their nanomorphology as well. Materials with similar morphological structures show different photocatalytic performances. TiO_2_ nanorods appear to have the best features of efficient photocatalyst, while the nanoflowers show very low efficiency. For CdS, the nanoflower is the best morphology for photocatalysis. It appears that high surface area is the key apart from the morphology, which controls the efficiency. The overall understanding by analyzing all the available information has enumerated a path to select an effective photocatalyst amongst the several nanomaterials available. Such an analysis and comparison is unique and has provided a handle to select the effective morphology of nanomaterials for photocatalytic applications.

## 1. Introduction

The nanotechnology industry has been envisioned to grow to the tune of 50 billion USD in the year 2022 [1]. The market survey done by Allied Market Research Group projected a compound annual growth rate (CAGR) of 20.7% from 2016 to 2022 for nanomaterials (NMs) that are widely used in industries [2]. It has become crucial for every country to invest in the development of NMs and their applications to keep pace with the growing technological advancement. This has happened because of the unique properties possessed by the devices based on patterns that one could engineer at atomic and molecular levels. Applications in nanotechnology are in wide-ranging fields like water purification, medicine, cosmetics, energy storage, healthcare, aerospace, consumer goods, agriculture, electronics, plastics, coatings, and electronic and electrical industries [3,4]. Due to the enormous need in the future for nanomaterials, it is crucial to understand the synthesis and growth of these nanostructures [5]. Besides considering the role of surface area, crystal structure, grain boundaries, chemical nature, density, and interfacial interaction of NMs with other materials, the properties of NMs are regulated to a large extent through the morphology of the NMs. 

According to Pokropivny and Skorokhod, NMs can be classified as 0D, 1D, 2D and 3D, respectively [6]. It was considered that the characteristics of the NMs depend on the shape of the particle and its dimensionality. Thus, the application of the NMs can be devised based on their morphology and dimensions, which control, to a large extent, the movement of electrons within the NMs. The term 0D is often used for the NMs in which the electrons are entrapped in the dimensionless space. In 1D type NMs, electrons can move along a particular direction, whereas in 2D and 3D type NMs, the electrons conduct in the plane and all directions, respectively. Various morphologies of NMs such as rods, spheres, flowers, wires, tubes, and belts have already been reported in the literature and also had a scope of different structures to be synthesized in future. Synthesis processes and the parameters such as time and temperature regulate the synthesis of different morphologies of NMs. On the other hand, different morphologies of the same nanomaterial display varying performance depending upon the type of application intended. Presently there are no reports/reviews which have focused entirely on this specific aspect of the role of varying morphologies of a nanomaterial concerning its synthesis and efficiency towards a specific application.

Environmental contamination is growing at an alarming rate due to rapid growth in population and industrialization. Organic pollutants are contaminating air, water and soil at catastrophic rates. Conventional technologies such as adsorption, ultrafiltration, coagulation and photocatalysis are routinely used for the decontamination of anthropogenic organic contaminants. The efficiency and economic cost of these technologies govern their wide-scale application in the detoxification of the environment. Consideration of efficiency in terms of the performance of the catalyst and time/energy required in its synthesis plays a vital role in its commercialization and application in the large-scale treatment of pollutants. The photocatalytic performance of NMs is the most researched topic in the present context. The past two decades have witnessed extensive research on semiconductor nanoparticles with unique photoinduced polarization properties and size-dependent band gap expansion. Due to their ability to oxidize the organic and inorganic substrates, these NMs find their use as photocatalysts for the removal of organic and inorganic pollutants either from the aqueous or gas phase. Compared to conventional chemical oxidation methods, semiconductor catalysts are low-cost, nontoxic oxides that do not lose photocatalytic activity on extended use. The photocatalytic activity of the semiconductor material is due to the generation of electrons/holes on the absorption of photons, which are influenced by a band gap. Photoexcitation of the electrons occurs when the incident energy of the photon equals or is larger than the band gap of the particular semiconductor catalyst. It triggers the photochemical reaction on the surface of the NMs. In this process, the performance of the semiconductor material is governed by optical absorption, charge separation and, finally, surface reactions. The morphology of the photocatalyst dramatically affects the efficiency of photocatalytic reactions. The unusual structural and optical properties of these NMs are controlled by the change in morphology, which in turn is governed by the synthesis process adopted, the time required for synthesis and reaction temperature. Hence, to employ the best morphology of a particular photocatalyst, these factors are required to be viewed together for process optimization. Interest in strategies to control particle morphology has increased in the past decade because of its potential in various applications. For the production of NMs with controlled morphology, many alternative methods like lyophilization, precipitation, hydrothermal, freeze-drying, spray-drying, emulsion-based, mechanical milling, and sol-gel methods have been proposed [7,8,9]. Of all these synthesis methods, Ghoderao et al. [10] found the hydrothermal method to be the most efficient synthetic method not only due to its simplicity and low-cost feasibility but due to the crystalline nature of the product obtained with a large surface of the photocatalyst resulting in a higher % of photodegradation. Nandiyanto et al. [11] reported that the aerosol-assisted self-assembly, besides being rapid and economically produced spherical particles that were free from agglomeration with a relatively monodispersed size. It was observed that a change in the process conditions could produce various particle morphologies. For instance, the spray method was used to generate spherical-shaped particles with the advantage of maximum structural stability [12].

Even though an impressive variation of morphologies ranging from spheres to rods, needles, cubes, hollow rods, flowers, square plates, ribbons and belts are available in the literature [13,14,15,16], in most of the studies, a clear correlation between the morphology and the photocatalytic efficiency is not adequately explained. Different NMs have been evaluated in terms of photocatalytic efficiency using different morphologies under different experimental degradation conditions; however, due to the absence of an elaborate description, clear reasoning for their activity may not be completely evident. This may be related to the difficulties in obtaining a well-defined surface and, consequently, a lack of clarity of the exposed crystallographic faces. Moreover, it is a tough task to compare various morphologies whose efficiencies are different due to different photocatalysis procedures adopted. Being a multi-step process, the photocatalytic degradation of a pollutant may impact differently depending on the relative reactivity of the exposed surfaces of NMs. For an efficient photocatalyst, electrons and holes must first be efficiently photogenerated within the bulk material, followed by their migration to the surface, where their surface trapping may depend on the crystalline quality and the shape of the particle morphology. However, it is also possible while migrating to the surface of the photocatalyst, the photoinduced electrons and holes might become inactive through recombination. A short distance from the core of the photocatalyst to its surface and a suitable concentration gradient reduces this recombination. The concentration gradient has a close correlation to the morphology and surface properties of the NMs [17].

The morphology of the semiconducting photocatalytic materials plays a vital role as the irradiated surface initiate reactions such as oxidation or reduction of the contaminants in the desired process. It is the relative reactivity of the exposed surfaces of NMs that directs further surface reactions directly with the pollutant to yield active radicals that are prone to degrade pollutants. It is, therefore, imperative to consider these parameters for selecting the catalyst amongst the different morphologies of the available photocatalytic materials. Semiconductor materials such as TiO_2_, ZnO, Ag_3_PO_4_, CdS and ZnTiO_3_ are emerging as some of the most popular materials for the treatment of contaminants by the photocatalytic process. It can be regarded as a potential material of the future required in large quantities. This is due to their environmental adaptability and ability to harness solar energy, which is abundantly available on earth for pollution control. Since the scope of metal oxide-based photocatalysts is broad, selective examples of the widely accepted photocatalysts such as TiO_2_, ZnO, Ag_3_PO_4_, CdS, and ZnTiO_3_ have been compared based on their photocatalytic performance, the time required for synthesis and reaction temperature. This review is significant for the energy and environment sectors, providing a perspective on the assessment of key photocatalysts, and will provide a direction to the material scientist, chemist and technologist working in the area of environmental mitigation.

## 2. Discussion on Various Nanomaterials

### 2.1. Titanium Dioxide (TiO_2_) Nanostructures

**Figure 1 molecules-27-07778-f001:**
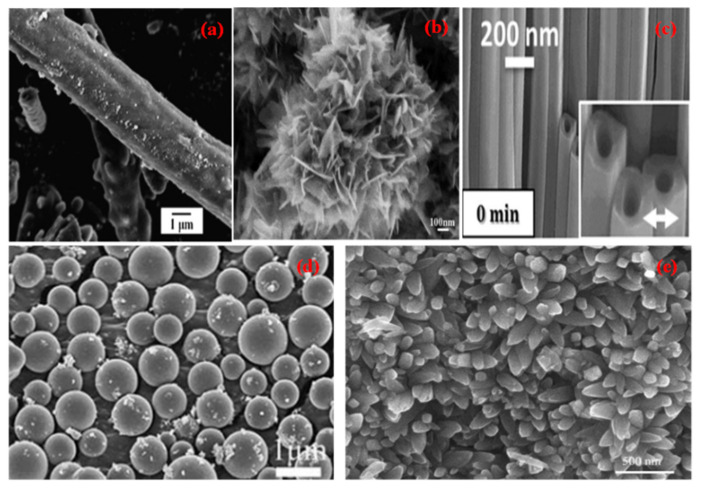
SEM micrographs of different nano morphologies of TiO_2_ (**a**) Nanorod (Reprinted with permission from [18]. Copyright 2014 Elsevier), (**b**) Nanoflower (Reprinted with permission from [19]. Copyright 2012 Elsevier), (**c**) Nanotubes (Reprinted with permission from [20]. Copyright 2014 Springer Nature), (**d**) Nanospheres (Reprinted with permission from [21]. Copyright 2011 Royal Society of Chemistry), (**e**) Nanospindles (Reprinted with permission from [22]. Copyright 2018 Elsevier).

Due to the nontoxic nature, natural and abundant availability, wide spectrum application range, and attractive physical properties, titanium dioxide-based nanomaterials of different morphologies (Figure 1) are of tremendous interest in the field of photocatalysis [23], photovoltaics [24], hydrogen production [25], self-cleaning coatings [26], energy storage [27], and antifogging coatings [28]. Different morphologies of titania NMs including nanorods, nano spindles, nanotubes, nanospheres, and nanoflowers, can be prepared using different approaches such as sol-gel method, electrochemical route, hydrothermal method, atomic layer deposition approach (ALD) etc. It becomes imperative to understand the effect of various morphologies of TiO_2_ nanomaterial on the surface physiochemical properties during its growth and end application. The performance of TiO_2_ as a photocatalyst primarily depends upon ion transport across the catalyst material, mass transfer to catalytic sites and charge transfer at the available surface of the material, and all of the above depends on the morphology of TiO_2_. The photocatalytic process is controlled to a large extent by the morphology of the titania nanomaterials and can be improved by the large surface area and nano-size of the nanoparticles [29]. It can ultimately control nature, density, crystallinity and defect concentration in the nanomaterials. It has a profound effect on charge transfer and ion transport properties. In contrast to the bulk titania phase, the morphology of nanostructures such as nanorods, nanoflowers, nano spindles, nanotubes, and nanospheres show high surface area and ease of mass transfer.

**Table 1 molecules-27-07778-t001:** Photocatalytic activity study of different nano morphologies of TiO_2_.

S. NO.	Material & Morphology	Method	Application	Photocatalytic Activity	Reference
		Nanorod			
1	TiO_2_/Nanorod	Sol-gel	*P*-nitrophenol, under 15 W UV Philips lamp	69% in 20 min	[30]
2	TiO_2_/Nanorod	Chemical vapour deposition	Methyl Orange (MO) & Methylene Blue (MB) under 100 W UV mercury lamp	97% of MO in 100 min 99% of MB in 50 min	[18]
3	TiO_2_/Nanorod	Sol-gel	Phenol under18 W UV lamp,	48% in 360 min	[23]
4	TiO_2_/Nanorod	Hydrothermal Method at 120 °C for 15 h	Phenol under UV light	87% in 360 min	[31]
5	TiO_2_/Nanorod	Hydrothermal Method at 200 °C for 12 h	MO under 300 W UV Xenon lamp	100% in 95 min	[32]
6	TiO_2_/Nanorod	Hydrothermal Method at 225 °C for 24 h	MO & MB under 6 W UV lamp	100% of MO & 88% of MB in 120 min	[33]
7	TiO_2_/Nanorod	Hydrothermal Method at 180 °C for 12 h	Phenol, 20 W UV lamp, 365	55% in 360 min	[21]
8	TiO_2_/rod	Hydrothermal, 200 °C for 18 h	MB, under 6 W UV lamp	80% in 100 min	[34]
9	TiO_2_/Nanorod	Hydrothermal Method at 180 °C for 24 h	MO, under UV mercury lamp	51% in 150 min	[35]
		Flower			
10	TiO_2_/Flower	Sol-gel	Phenol, under 18 W UV lamp	70% within 360 min	[23]
11	TiO_2_/Flower	Hydrothermal, 180 °C for 12 h	Phenol, 20 W UV lamp	97% in120 min	[21]
12	TiO_2_/Flower	Hydrothermal, 180 °C for 6 h	Rhodamine B (RhB), under 450 W UV Xenon lamp	69% in 160 min	[36]
13	TiO_2_/Flower	Sol-gel	RhB under 300 W UV lamp	91.4% in 50 min	[37]
14	TiO_2_/Flower	Hydrothermal, 150 °C for 24 h	RhB, under 350 W Xenon Visible lamp	63% in 60 min	[38]
15	TiO_2_/Flower	Hydrothermal, 150 °C for 3 h	MB under UV lamp	78% in 60 min	[39]
16	TiO_2_/Flower	Hydrothermal, 120 °C for 48 h	MO under sunlight	60% in 60 min	[19]
17	TiO_2_/Flower	Hydrothermal, 150 °C for 24 h	MB under UV 300 W high-pressure mercury (Hg) lamp	75% in 60 min	[40]
		Tube			
18	TiO_2_/Tube	Furnace 500 °C for 4 h	Papermaking wastewater, under 375 W high-pressure Hg lamp	99.5% in 720 min	[41]
19	TiO_2_/Tube	Electrochemical Method	MB, under UV lamp	98% in 60 min	[20]
20	TiO_2_/Tube	Hydrothermal, 160 °C for 24 h	MO, under 300 W UV lamp	50.2% in 60 min	[42]
21	TiO_2_/Tube	Sol-gel stirring at 40 °C for 24 h	RhB & Dibutyl phthalate (DBP) 125 W high-pressure Hg UV lamp	20% of RhB in 60 min & 15% of DBP in 60 min	[43]
22	TiO_2_/Tube	Solvothermal, 180 °C for 24 h	Orange II, under 18 high-pressure Hg lamps	97.98% in 3000 min	[44]
23	TiO_2_/Tube	Electrochemical method	MB, under UV light lamp	72% in 200 min	[45]
24	TiO_2_/Tube	Electrochemical Method	Phenol, under 1000 W Xenon lamp visible light lamp	99.5% in 40 min	[46]
		Sphere			
25	TiO_2_/Sphere	Hydrothermal, 180 °C for 24 h	Phenol, under 20 W UV lamp,	60% in 120 min	[21]
26	TiO_2_/Sphere	Hydrothermal, 130 °C for 48 h	MB, under UV lamp	96% in 80 min	[47]
27	TiO_2_/Sphere	Hydrothermal, 200 °C for 18 h	MB, under 6 W UV lamp	90% in 100 min	[34]
28	TiO_2_/Sphere	Hydrothermal, 80 °C for 24 h	MB, under UV light	96% in 100 min	[48]
29	TiO_2_/Sphere	Hydrothermal, 160 °C for 24 h	MO, under 4 W UV lamp	50% in 60 min	[49]
30	TiO_2_/Sphere	Hydrothermal, 150 °C for 72 h	MO, under 8 W UV lamp	91.6% in 60 min	[50]
		Spindle			
31	TiO_2_/spindle	Hydrothermal, 180 °C for 12 h	MO, under 300 W visible light	38% in 120 min,	[51]
32	TiO_2_/spindle	Hydrothermal, 200 °C for 24 h	RhB, under 350 W Xenon visible lamp	23% in 60 min	[52]
33	TiO_2_/spindle	Reverse micellar method	RhB, under UV lamp	90% in 130 min	[53]
34	TiO_2_/spindle	Hydrothermal, 180 °C for 12 h	MO, under 250 W UV high-voltage Hg lamp	91% in 300 min	[54]
35	TiO_2_/spindle	Hydrothermal, 200 °C for 24 h	RhB, visible light	25% in 60 min	[55]

Various morphologies of TiO_2_ were synthesized using different approaches (Table 1), such as sol-gel [56,57], hydrothermal, electrochemical, solvothermal and solid-state methods. The nanorods were synthesized using sol-gel [30], solid-state [58], and hydrothermal methods [34]. Generally, it takes two hours for the complete mineralization of organic pollutants using TiO_2_ nanorods. TiO_2_ nanoflowers were prepared by using the sol-gel [34] and hydrothermal methods [21,36]. It takes more than an hour to remove organic contaminants by a photocatalytic process using TiO_2_ nanoflowers. The nanotubes were prepared by using electrochemical, hydrothermal [59], sol-gel [60], and solvothermal methods [61]. The 99.5% removal of organic pollutants was achieved within 40 min using TiO_2_ nanotubes [46]. TiO_2_ nanospheres were synthesized mostly by hydrothermal technique. It takes more than 2 h for the total mineralization of organic contaminants photocatalytically by using TiO_2_ nanospheres. TiO_2_ nano spindles were obtained by using hydrothermal [52] and reverse micellar [20] approaches. Nanospindle morphology was seen to be the slowest among all morphologies for photocatalytic mineralization of organic pollutants. It takes more than two hours to complete the mineralization of organic contaminants by using TiO_2_ nano spindles.

Therefore, it is important to study the parameters involved, such as surface area, the temperature of crystal growth, the time required for synthesis as well as photocatalytic activity in unison and collectively along with the range of different morphologies for a holistic understanding. The data collected from existing literature indicates that on all counts, TiO_2_ nanotube morphology proved to be excellent amongst different morphologies compared, as stated above. We have considered the best results from the available literature for comparison and analysis (Figure 2).

It can be observed from Figure 2 that different morphologies can be compared based on various parameters. The temperature of synthesis varies for obtaining different morphologies of TiO_2_. Nanorods, nanospindle, and nanospheres show the highest temperature requirement (200 °C), whereas nanoflower and nanotubes can be synthesized at a twenty-degree lower temperature. In the case of photocatalytic performance, nanorods, nanotubes, and nanospheres show the highest performance, whereas nanoflowers and nanospindle show a 10% lesser performance. The catalytic activity time taken by nanospindle is the highest, and nanotubes show the lowest among the reported. The nanorod and nanoflower show moderate time, but nanosphere time is a little higher comparatively. The surface area of nanotubes and nanospheres is the highest among the reported, whereas nanorods and nanospindle show a 50% lesser surface area as compared to nanotubes and nanospheres. The surface area of nanoflower shows a moderate value comparatively.

The remarkable catalytic activity (99%) with minimum activity time (40 min) was observed in the case of TiO_2_ with nanotube morphology [46]. This can be related to the high surface area (261 m^2^/g) and ease of access to the reactants during the catalytic process. Thus, it can be inferred that surface area and easy access to reactants are essential factors which govern the performance of the different morphologies in TiO_2_. The increased surface area makes available microchannels for the free flow of reactants and degradation products during the photocatalytic process. These microchannel increases the possibility of interaction with a more catalytic active center, thereby increasing the performance of the catalyst. The highlight of this tube-like morphology is its low-temperature synthesis (180 °C) by solvothermal route [61]. The different morphologies of TiO_2,_ such as nanorods and nanospheres, also show excellent performance in terms of photocatalytic activity but have less surface area and more activity time as compared to the nanotube morphology. However, the efficiency of nanospheres is close to nanotubes and can be considered as the next best material matched with nanotubes [62]. Further, nanoflower morphology with a high surface area lacks the desired catalytic performance, probably due to poor accessibility of reactive species to active centers during the photochemical process [23]. The different morphologies studied in decreasing order of their efficiency are nanotubes [47], nanospheres [48], nanorods [18], nanospindle [53] and nanoflowers [37].

### 2.2. Zinc Oxide (ZnO) Nanostructures

**Figure 3 molecules-27-07778-f003:**
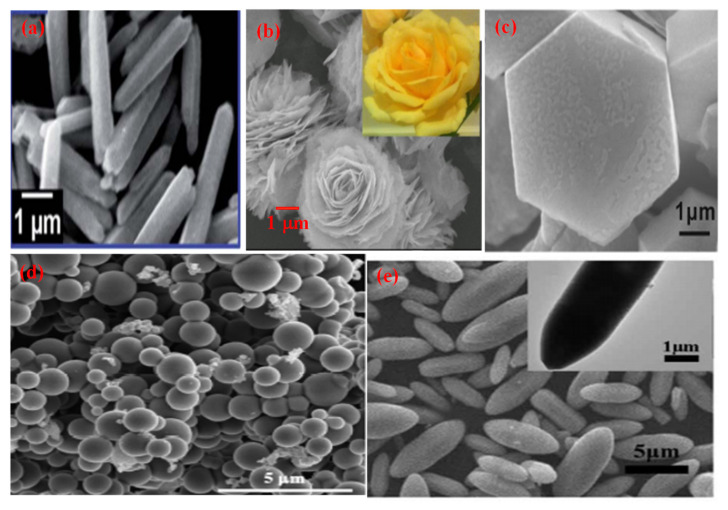
SEM micrographs of different morphologies of ZnO (**a**) Nanorod (Reprinted with permission from [63]. Copyright 2014 Royal Society of Chemistry), (**b**) Nanoflower (Reprinted with permission from [64]. Copyright 2016 Elsevier), (**c**) Hexagonal (Reprinted with permission from [65]. Copyright 2017 Royal Society of Chemistry), (**d**) Nanospheres (Reprinted with permission from [66]. Copyright 2019 Elsevier), (**e**) Nanospindles (Reprinted with permission from [67] Copyright 2017 Elsevier).

Different morphologies of zinc oxide (ZnO) are (Figure 3) considered a vital semiconductor catalyst with a wide band gap of 3.37 eV with high excitation binding energy of 60 eV and have been studied extensively for photocatalytic reactions. The performance of ZnO as a semiconductor photocatalyst can be related to the absorption of light, which excites electrons from the valence band (VB) to the conduction band, which creates a hole (h^+^) in VB and initiates the photo redox reaction. Zinc oxide occurs in nature as a zincite mineral which has a hexagonal structure with lattice parameters a and c of 3.25 Å and 5.20 Å, respectively. It is used for making pigments, transducers, lasers, diodes, sensors and catalysts. The most common use of ZnO in the nanoscale range is for making sunscreen additives for protection from UV radiation, and ZnO thin films are used in making devices. Thus, ZnO can be used in a broad range of applications starting with catalytic activity to optoelectronics and to antimicrobial activity as well, which leads to a variety of products manufactured. ZnO is preferred as a semiconductor catalyst due to its low cost, efficiency, nontoxic nature, high redox potential and ease of availability. When compared to TiO_2,_ the photocatalytic performance of ZnO is much superior. It is due to the absorption of a more significant portion of the UV spectrum by ZnO with greater electron mobility (200–300 cm^2^V^−1^s^−1^) and higher oxidation potential of the ^•^OH radical generated [68]. This results in a rapid electron transfer leading to a higher quantum yield. ZnO has a higher recombination rate of electrons and holes generated during photocatalysis which reduces its performance as a photocatalyst.

**Table 2 molecules-27-07778-t002:** Comparative study of different nano morphologies of nanostructured ZnO.

S. NO.	Material & Morphology	Method	Application	Performance	Reference
		Rod			
1	ZnO/Rod	Microwave reactor (heated to 80 °C for 10 min)	MO, under 300 W Hg lamp	86.3% in 180 min	[69]
2	ZnO/Rod	Hydrothermal, 180 °C for 24 h	Resorcinol, under 15 W UV lamp	100% in 120 min	[70]
3	ZnO/Rod	Atmospheric self-induction method	RhB, under 400 W Xenon visible lamp	36.8% in 300 min	[71]
4	ZnO/Rod	Solvothermal, 80 °C for 5 h	MB under 300 W UV lamp	100% in 20 min	[72]
5	ZnO/Rod	Hydrothermal, 140 °C for 12 h	MB under 6 W UV lamp	98.5% in 100 min	[73]
6	ZnO/Rod	Hydrothermal, 120 °C for 20 h	Phenol under l 15 W UV lamp	100% in 40 min	[74]
7	ZnO/Rod	Hydrothermal, 95 °C for 30 h	RhB under 500 W visible Xenon lamp	50% in 300 min	[75]
		Flower			
8	ZnO/Flower	Hydrothermal, 100 °C for 12 h	RhB, under 300 W Hg lamp	99.84% in 25 min	[64]
9	ZnO/Flower	Thermal decomposition at 300 °C for 20 min	RhB, under 36 W UV lamp	100% in 90 min	[76]
10	ZnO/Flower	Sol-gel at 80 °C	RhB, under 200 W high-pressure Hg UV lamp	99.8% in 100 min	[77]
11	ZnO/Flower	Hydrothermal, 140 °C for 12 h	MB, under 6 W UV lamp	94% in 100 min	[73]
12	ZnO/Flower	Precipitation method	RhB, under Hg UV lamp	30% in 180 min	[78]
13	ZnO/Flower	Hydrothermal, at 190 °C for 1 h	MB, 125 W Hg UV lamp	98% in 60 min	[77]
14	ZnO/Flower	Solution based at 97 °C for 4 h.	MB, under 30 W Hg UV lamp	99.9% in 180 min	[79]
15	ZnO/Flower	Microwave, at 300 W for 12 s	MB, under high-pressure Hg UV lamp	80% in 60 min	[63]
16	ZnO/Flower	Hydrothermal, at 90 ° C for 24 h	MB, under BLB UV lamp	100% in 105 min	[80]
17	ZnO/Flower	Sol-gel at room temperature for 16 h.	RhB, under 300 W Xenon UV lamp	100% in 100 min	[81]
		Sphere			
18	ZnO/Sphere	Hydrothermal, at 140 °C for 12 h	MB, under 6 W UV lamp	74% in 100 min	[73]
19	ZnO/Sphere	Hydrothermal, at 180 °C for 24 h	RhB, under 15 W UV lamp	100% in 240 min	[82]
20	ZnO/Sphere	Heated in a silicone bath at 120−140 °C for 4 h	MO, under 24 W UV lamp	90% in 300 min	[83]
21	ZnO/Sphere	Hydrothermal, at 180 °C for 4 h	Congo Red, under 30 W UV lamp	99.2%, in 90 min	[84]
22	ZnO/Sphere	Hydrothermal, at 120 °C for 6 h	MB, under 80 W UV lamp	95% in 60 min	[85]
		Hexagonal			
23	ZnO/Hexagonal	Heated at 150 °C on a hotplate	MB, under 450 W medium pressure Hg UV lamp	100% in 16 min	[68]
24	ZnO/Hexagonal	Calcined at 400 °C	MB, under 16 W UV lamp	100% in 75 min	[86]
25	ZnO/Hexagonal	Solvothermal, at 110 °C for 10 h	RhB, under Hg UV lamp	80% in 60 min	[87]
26	ZnO/Hexagonal	Sol-gel at 80 °C for 3 h	MB, under 100 W UV lamp	100% in 20 min	[88]
27	ZnO/Hexagonal	Hydrothermal, 120 °C for 20 h	MB, under UV lamp	100% in 60 min	[74]
28	ZnO/Hexagonal	solid-phase method	MO, under 300 W UV lamp	96.4% in 60 min	[89]
29	ZnO/Hexagonal	Hydrothermal, at 200 °C for 24 h	MB, under 300 W Hg UV lamp	60% in 180 min	[90]
30	ZnO/Hexagonal	Sol-gel at 80 °C for 12 h	MB, under UV lamp	95% in 60 min	[65]
31	ZnO/Hexagonal	Sonochemical method	MB, under 400 W Xenon UV lamp	97% in 30 min	[91]
		Spindle			
32	ZnO/Spindle	Hydrothermal, 140 °C for 12 h	MB, under 365 UV lamp	62% in 100 min	[73]
33	ZnO/Spindle	Hydrothermal, 120 °C for 8 h	MO, under UV lamp	55% in 180 min	[67]
34	ZnO/Spindle	Hydrothermal, 150 °C for 3 h	RhB, under 8W HG UV lamp	73% in 120 min	[92]
35	ZnO/Spindle	Microwave, at 110 °C for 17 min	MB, under 300 W high-pressure Hg UV lamp	98% in 120 min	[93]
36	ZnO/Spindle	Hydrothermal, at 140 °C for 12 h	MB, under 6 W high-pressure Hg UV lamp	72% in 100 min	[73]
37	ZnO/Spindle	Hydrothermal, 95 °C for 24 h	MB, under 60 W Hg UV lamp	83% in 55 min	[94]

There are different methods, such as hydrothermal, sol-gel, coprecipitation, microwave, sonochemical and solid-state approaches which have been used in the preparation of zinc oxide with varying morphologies (Table 2). Zinc oxide nanorods were synthesized using microwave irradiation [69], hydrothermal [73,75] and solvothermal methods [72]. The average time taken for complete mineralization is around two hours. Zinc oxide nanoflowers were prepared using hydrothermal [64,73,77,80], sol-gel [79,81,95], coprecipitation [78], and microwave irradiation method [63]. It is reported to take about two hours for the nanoflower morphology to completely remove organic contaminants photocatalytically from an aqueous medium. Zinc oxide nanospheres were synthesized mostly using the hydrothermal route, and when applied for environmental photocatalysis, it takes less than 2 h for the complete mineralization of organic contaminants from water. The hexagonal morphology of zinc oxide was made using solvothermal [87], sol-gel [65,88,96], hydrothermal [90], solid-state [89] and sonochemical [91] approaches. It takes less than 1 h for the removal of organic pollutants by using hexagonal morphology. Nanospindle morphology was prepared mostly using hydrothermal routes [72,94]; however, the microwave technique [93] was also applied in its synthesis. The complete mineralization of organic contaminants achieved by using nano spindles was less than two hours.

Thus, it will be quite interesting to know as to which factors are responsible for the best performance of ZnO morphologies as a photocatalyst. ZnO nanomaterials show different morphologies such as nanorods [72], nanoflowers [97], nanospheres [85], nanospindle [94] and hexagonal morphology [91]. To analyze these different morphologies, the best results available in the literature were compared (Figure 4).

Figure 4 indicates that the nanorods, nanospheres and hexagonal morphology of ZnO required more time for synthesis (24 h), whereas nanoflower and nanospindle morphologies need 50% lesser time, comparatively. The temperature needed for the synthesis of hexagonal, nanorod and nanospheres is maximum (200 °C); however, nanoflower and nanospindle morphology require thirty per cent lesser temperature relatively. In the case of photocatalytic performance, nanorod, nanoflower, and hexagonal morphologies show the highest value, while the performance of nanospindle and nanospheres shows a little lesser value comparatively. The photocatalytic activity time taken by nanospheres and nanospindle morphology is the highest amongst reported morphologies, while nanorods, nanoflowers and hexagonal morphology show moderate values. The surface area of nanoflowers shows the highest values, whereas nanorods, nanospheres, and hexagons have modest values, but nanospindle reveal the lowest among the reported morphologies.

It is observed that ZnO nanoflower morphology shows the best results [98] in terms of surface area, time of synthesis, time of catalytic activity, and performance, amongst all other reported morphologies. Figure 3b shows the highest surface area (96 m^2^/g) for nanoflower morphology which has a direct impact on the performance of the catalyst. The surface area plays a vital role in mass transfer and charge transfer on the active catalytic surface. The time required (12 h) for the synthesis of nanoflower is also the lowest among other reported morphologies. The nanoflower morphology shows the removal of pollutants within 25 min, and the temperature of synthesis is the lowest compared to other morphologies. Thus, the nanoflower morphology of the ZnO stands out to be the best photocatalyst on all counts as equated with other reported morphologies of ZnO. The other morphology next to nanoflower is hexagonal in terms of surface area (56 m^2^/g), which enables the complete removal of pollutants within 16 min by the photocatalytic route. However, considering the time for synthesis of (24 h at 200 °C) of the hexagonal structure needs further optimization [68]. The nanorod morphology of ZnO shows complete removal of the pollutants within 20 min, whereas, in the case of nanosphere morphology, it takes an hour to achieve the same. The nano spindle morphology shows poor results in terms of catalytic performance (83%) and time taken to remove pollutants. However, the good thing is that the time and temperature of synthesis, i.e., 12 h and 140 °C, respectively, are more economical. Thus, the order of the morphologies for ZnO in terms of significance may be considered (from most significant to the lowest) as nanoflowers, nanospheres, nanorods, nano hexagons and lastly, nanospindle.

### 2.3. Cadmium Sulphide (CdS) Nanostructures

**Figure 5 molecules-27-07778-f005:**
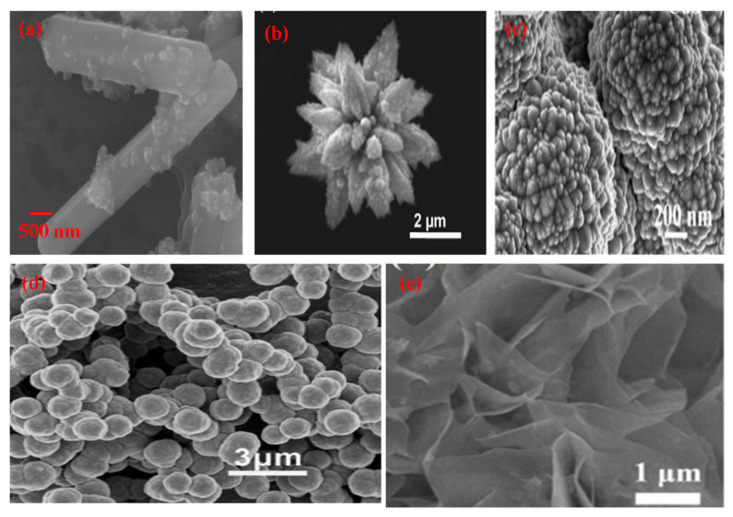
SEM micrographs of different morphologies of CdS (**a**) Nanorod (Reprinted with permission from [99]. Copyright 2018 Elsevier), (**b**) Nanoflower (Reprinted with permission from [100]. Copyright 2017 Elsevier), (**c**) Particles (Reprinted with permission from [101]. Copyright 2019 Elsevier), (**d**) Nanosphere (Reprinted with permission from [102]. Copyright 2013 Elsevier), (**e**) Sheet (Reprinted with permission from [103]. Copyright 2018 Elsevier).

Synthesis of various morphologies of CdS nanostructures has been achieved by employing different approaches such as the solvothermal method, hydrothermal method, irradiation technique, electrodeposition process, liquid crystal template route [104] etc. Systematic research on the influence of phase behaviour has opened up new avenues for the generation of different morphologies, such as nanorods, nanospheres, and nanoflowers. According to Onsager [105], at low volume fractions, nanorods can form a distinct phase of anisotropic crystals having the positional disorder and leading to a nematic phase. Entropy drives the phase transition in the synthesis of such morphology. The contact of the photocatalyst surface with reactants has a remarkable influence on photocatalytic reactions, and the interfacial contact depends upon the morphology of the catalyst and the surface area available. In the case of CdS, nanomaterials morphology and particle size have an enormous impact on the photocatalytic property, and hence researchers are working on controlling the morphology of CdS nanomaterials to enhance the photocatalytic activity. Different morphologies such as nanosheets [103], nanorods [106], nanospheres [107], nano particles [108], and nanoflower [109] have been explored in this context (Figure 5). Several synthetic routes, such as chemical vapor deposition, thermal deposition, electrodeposition, template approach, microwave-assisted process, and solvothermal and hydrothermal methods, were applied to achieve the desired morphology of CdS nanomaterials. Out of these synthesis approaches hydrothermal route was found to be the most preferred and yielded the best results for optimal morphology. CdS has a narrow and direct band gap of 2.42 eV and can be considered an ideal II-VI semiconductor photocatalyst in the visible region. Thus, it is the most widely studied photocatalyst for several photocatalytic transformations, such as water splitting, reduction of nitro compounds, dye degradation, and photoelectric conversions. Tri-n-octyl phosphine oxide (TOPO) was used by Alivisatos [110] as a size and shape-controlling ligand for obtaining CdS nanomaterials. CdS nanowires were synthesized by Lieber et al. [111]., by using a laser ablation technique. Nanorods, nanowires and nanotubes were synthesized by using a thermal approach or a CVD method. The Ethylenediamine template approach was used [112] in the synthesis of nanorods and nanowires via a solvothermal route.

The Nanoflower morphology of semiconductor materials (CdS) is being applied as a chemo-sensor, photocatalyst, and catalyst for microbiological processes. As compared to other materials like ZnS, ZnO, TiO_2_, CdSe and similar other semiconductor materials, CdS has shown an increase in photocatalytic activity under visible light irradiation due to its narrow bandgap of 2.4 eV [113]. The nanomorphology and surface area are mainly responsible for its light absorption ability and charge carrier’s characteristics. Further, the ordered flower-like microstructure also boosts the photocatalytic properties of CdS. The nucleation and growth of different shapes of nanoflowers decide the order of construction of CdS nanoflowers. The actual morphology of the CdS microstructure can be tuned by adding surfactants or capping agents. The active surface of nanocrystals, due to the addition of surfactants, reduces interfacial activation energy and induces secondary nucleation during synthesis. There are many methods available in the literature for the synthesis of nanoflowers, such as rose-like flowers, nanocomposite flowers, petal-like structures and spindles. The CdS nanoflower morphology is applicable over a wide range of wavelengths and can be useful for the degradation of dyes like Rhodamine B and methylene blue under visible light. The structure of CdS nanoflowers contains more oxygen vacancies than that of other morphologies. The hydrothermal method is the most preferred in the synthesis of hierarchical CdS nanostructures due to its cost-effectiveness, scale-up, easy control and better crystalline quality of the product obtained. Hierarchical morphologies of CdS play an essential role during photocatalytic activity. It enhances the performance of the photocatalyst by increasing the absorption efficiency of the incident radiation and further transferring the energy to reactant molecules [114]. The charge separation in the hierarchical structure is much more efficient. To attain the maximum efficiency of CdS nanostructures, the synthesis of controllable hierarchical nano morphologies become important.

**Table 3 molecules-27-07778-t003:** Details of the synthesis and photocatalytic performance of various morphologies of CdS.

S. NO.	Material & Morphology	Method	Application	Performance	Reference
		Rod			
1	CdS/Rod	Hydrothermal, 180 °C for 6 h	MB, under 300 W Xenon visible lamp	70% in 80 min	[115]
2	CdS/Rod	Reflux method for 13 h	MB, under 300 W Xenon visible lamp	95% in 50 min	[106]
3	CdS/Rod	Hydrothermal, 160 °C for 48 h	Malachite green (MG) & MO, under 300 W Xenon visible lamp	67% of MG in 30 min & 58% of MO in 45 min	[116]
4	CdS/Rod	Hydrothermal, 120 °C for 10 h	Salicylic acid and *p*-nitrophenol under 125 W Hg UV lamp	70% Salicylic acid & 43.7% p-nitrophenol in 240 min	[117]
5	CdS/Rod	Hydrothermal, 180 °C for 12 h	Congo red (CR), under a visible tungsten lamp	40% in 25 min	[118]
6	CdS/Rod	Hydrothermal, 180 °C for 24 h	MB, under a 100 W visible lamp	62% in 180 min	[119]
7	CdS/Rod	Hydrothermal, 160 °C for 12 h	MB, under Hg UV lamp	35% in 120 min	[114]
8	CdS/Rod	Hydrothermal, 160 °C for 24 h	Cr (VI), under a 1 kW Xenon visible light lamp	19% in 120 min	[120]
9	CdS/Rod	400 °C for 1 h in an N_2_ atmosphere	RhB, under a 200 W tungsten halogen visible lamp	100% in 55 min	[121]
10	CdS/Rod	Hydrothermal, 200 °C for 10 h	MB, under a Xenon visible lamp	50% in 120 min	[122]
11	CdS/Rod	Hydrothermal, 180 °C for 1 h	Ciprofloxacin (CIP), under a 300 W Xenon visible lamp	57% in 60 min	[123]
12	CdS/Rod	Wet chemical method under reflux condition (100 °C for 7 h)	MO under a 300 W UV mercury lamp	93% within 40 min	[124]
		Flower			
13	CdS/Flower	Hydrothermal 200 °C for 12 h	MB, under a 125 W Hg visible lamp	100% in 220 min	[125]
14	CdS/Flower	Sol-gel method	MB, under a 300 W Xenon visible lamp	80% in 60 min	[126]
15	CdS/Flower	Hydrothermal 260 °C for 12 h	RhB, under a 300 W Xenon visible lamp	70%, in 180 min	[127]
16	CdS/Flower	Hydrothermal, 180 °C for 12 h,	Acid fuchsine, under a 125 W Hg UV lamp	100% in 40 min	[109]
17	CdS/Flower	Hydrothermal, 160 °C for 12 h	MB, under a 300 W Xenon visible lamp	100% in 180 min	[128]
18	CdS/Flower	Hydrothermal, 160 °C for 4 h	MB, MO & RhB, under a 500 W Xenon visible lamp	100% of MB, 91% of MO, and 85% of RhB in 150 min	[113]
19	CdS/Flower	Hydrothermal, 200 °C for 5 h	RhB, under visible light irradiation	93% in 120 min	[100]
		Sheet			
20	CdS/Sheet	Hydrothermal, 80 °C for 72 h	H_2_ production under a visible-light, AM 1.5 G solar simulator	20 μmol within 480 min	[129]
21	CdS/Sheet	Electrochemical deposition for 15 min	CO_2_ reduction under sunlight	2.1 μmol/g of C2H5OH and 62.8 μmol/g of HCOOH, 0.25% in 300 min	[130]
22	CdS/Sheet	Microwave method, at 80 °C for 30 min	H_2_ production, under visible light irradiation	27.4 μmol/g in 240 min	[131]
23	CdS/Sheet	Heated in an oil bath at 60 °C for 3 h	H_2_ production under a 350 W Xenon visible lamp	582 μmol/g in 240 min	[103]
24	CdS/Sheet	Ultrasonication at 90 °C for 2.5 h	RhB, under a 500 W Xenon visible lamp	50% in 180 min	[112]
		Sphere			
25	CdS/Sphere	Hydrothermal, at 120 °C for 10 h	Salicylic acid & p-nitrophenol, under a 125 W Hg UV lamp	20% Salicylic acid & 6.25% p-nitrophenol in 240 min	[117]
26	CdS/Sphere	Hydrothermal, at 160 °C for 12 h	MB, under a Hg UV lamp	38% in 120 min	[114]
27	CdS/Sphere	Hydrothermal, 180 °C for 4 h	Eosin Y, under a 500 W iodine tungsten lamp	100% in 120 min	[107]
28	CdS/Sphere	Ultrasonic method	MB, under a 125 W UV lamp	87% in 90 min	[132]
29	CdS/Sphere	Hydrothermal, at 100 °C for 2 h	4-Chlorophenol, under 65 W fluorescent visible lamps	52% in 150 min	[133]
30	CdS/Sphere	Microwave for 30 min	MB & RhB, under a 300 W Xenon visible lamp	95% of MB in 150 min, 90% of RhB in 180 min	[102]
31	CdS/Sphere	Hydrothermal, at 200 °C for 3.5 h	RhB, under a 300 W tungsten halide visible lamp	90% in 180 min	[134]
		Particles			
32	CdS/Particle	Hydrothermal, at 160 °C for 12 h	MO, under a 350 W Xenon visible lamp	12% in 60 min	[101]
33	CdS/Particle	Microwave for 20 s	Selective oxidation of alcohols to corresponding aldehydes under a 300 W Xenon visible lamp	94% in 60 min	[108]
34	CdS/Particle	Hydrothermal, at 160 °C for 12 h	MB, under a Hg UV lamp	29% in 120 min	[114]
35	CdS/Particle	Heating at 120 °C in an N_2_ environment	RhB, MB, & Cr (VI), under a 300 W Xenon visible lamp	21% of RhB, 16% of Cr (VI) in 20 min & 24% of MB in 40 min	[135]
36	CdS/Particle	Hydrothermal, at 160 °C for 24 h	RhB, under a 250 W visible lamp	72% in 240 min	[136]
37	CdS/Particle	Sol-gel method	MB, under a 300 W Xenon visible lamp	48% in 60 min	[126]

Herein, different morphologies such as nanorods, nanoflowers, nanosheets, nanospheres and nanoparticles of CdS have been reviewed. From Table 3, it can be observed that CdS nanorods were mainly obtained by the hydrothermal method. The time required in the synthesis by hydrothermal route varies from 1 h to 48 h. It was later applied in the mineralisation of organic pollutants and was reported to take an hour for the complete degradation of organic dye [121]. Generally, the temperature for the hydrothermal synthesis of the nanorods varies between 120 and 180 °C. CdS nanoflowers can be synthesised by using a hydrothermal route (most commonly used) where synthesis time and temperature vary between 4–12 h and 160–260 °C, respectively. The removal of organic dye photo catalytically was done within less than an hour [109]. The CdS nanosheets have been prepared by different methods, such as hydrothermal [129], electrochemical [130], microwave [131] and ultrasonication methods. It is reported that CdS nanosheets require three hours for the complete degradation of organic pollutants [112]. CdS nanosheets can also be applied in hydrogen production and CO_2_ reduction. Synthesis of CdS nanospheres was reported mostly by using hydrothermal routes [134], whereas other methods such as ultrasonication [132] and microwave irradiation [102] were also used. The removal of organic pollutants was carried out within three hours photocatalytically by using CdS nanospheres. CdS nanoparticles were synthesized by hydrothermal method [136], microwave technique [108] and sol-gel method [126]. The CdS nanoparticles required around 1 h for the complete degradation of organic dye [108].

CdS nanoparticles have been stabilized in different nano morphologies such as nanorods, nanoflowers, nanosheets, nanospheres and nanoparticles etc. We need to understand the morphology and its efficiency in terms of its end application and the time taken for photocatalytic activity, the time required for synthesis and the temperature of catalyst synthesis.

CdS nanosheet morphology requires 72 h for synthesis; however, nanorods, nanoflowers, and nanoparticles can be prepared in 17 h, and nanosphere morphology takes only 6 h for synthesis. For nanoflowers, nanospheres and nanorods need the highest reported temperature for synthesis, while nanoparticles can be prepared with a twenty per cent lesser value, and nanosheets can be synthesized at the lowest reported value of temperature. The photocatalytic performance of nanorods, nanoflowers, nanospheres and nanoparticles are the highest amongst those reported, while nanosheet shows a 50% lower value, comparatively. The time taken for the degradation of organic compounds photo catalytically is the maximum for nanosheet morphology; however other morphologies show an average of 1 h. The surface area of nanorods and nanoflowers is the highest among the reported, whereas nanosheets, nanospheres, and nanoparticles show moderate values.

It can be seen from Figure 6 that the surface area of CdS nanoflower (47 m^2^/g) is the highest among all the reported morphologies. The time required for the complete degradation of organic pollutants is also comparatively less than (40 min). It can be observed that the temperature of synthesis of various morphologies varies between 160 to 200 °C. In the case of CdS nanosheets, nanospheres, and nanoparticles, the reported surface area are 28, 18, and 19 m^2^/g, respectively. This seems to be the primary reason for the poor performance in terms of catalytic activity for these morphologies as compared to nanoflowers. Thus, based on the assessment parameters such as surface area, activity time, and photocatalytic performance, it can be seen that nanoflower morphology emerges as the best CdS nanostructure to be used as a photocatalyst. The order of preference of the different CdS morphologies for photocatalysis can be assigned as nanoflowers, nanorods, nanosheets, nanospheres and nanoparticles.

### 2.4. Silver Phosphate (Ag_3_PO_4_) Nanostructures

The treatment of pollutants in wastewater by heterogeneous photocatalysis in the presence of semiconductors has been regarded as a promising and efficient approach due to the high efficiency, eco-friendly nature, and easy recycling carried out under mild conditions. Among various photoactive materials, silver phosphate (Ag_3_PO_4_) nanostructure with different morphologies (Figure 7) has attracted much attention and has been found to be a fascinating material. It is an excellent photocatalyst in the visible region and has a high quantum efficiency [137] of about 90%. It has superior semiconductor properties for direct water splitting and photodecomposition of organic dyes [138]. It has been reported that the direct band gap of Ag_3_PO_4_ is 2.43 eV which can absorb wavelengths up to 530 nm [139]. The photocatalytic activity of Ag_3_PO_4_ depends upon its electronic structure, and its valence band (VB) is made up of 2p orbitals of oxygen and 4d orbitals of silver. However, the conduction band (CB) is made up of hybridized 5s and 5p orbitals of silver favoring electron transfer and energy dispersion in all directions. The presence of d states in the conduction band in other semiconductors increases electron-hole recombination and also hinders the mobility of electrons, resulting in a reduction in photocatalytic activity. However, here in the case of Ag_3_PO_4,_ the effect of d orbital in CB is inhibited. Therefore, the high photocatalytic activity is also attributed to the absence of d orbitals in the CB of Ag_3_PO_4_. However, the practical application of Ag_3_PO_4_ is still not satisfactory due to the formation of Ag^0^ particles on the surface of the photocatalyst during the photocatalysis process. Recently, some studies revealed that the photocatalytic activity of Ag_3_PO_4_ is directly influenced by size, morphology and presence of highly reactive facets [140]. To understand the properties of Ag_3_PO_4_ semiconductors, the surface and interfaces are crucial for photocatalytic activity. Hence, significant attention has been paid to the synthesis of morphology-controlled Ag_3_PO_4_, including cubes, dodecahedrons, tetrahedrons, spheres and polyhedrons.

**Figure 7 molecules-27-07778-f007:**
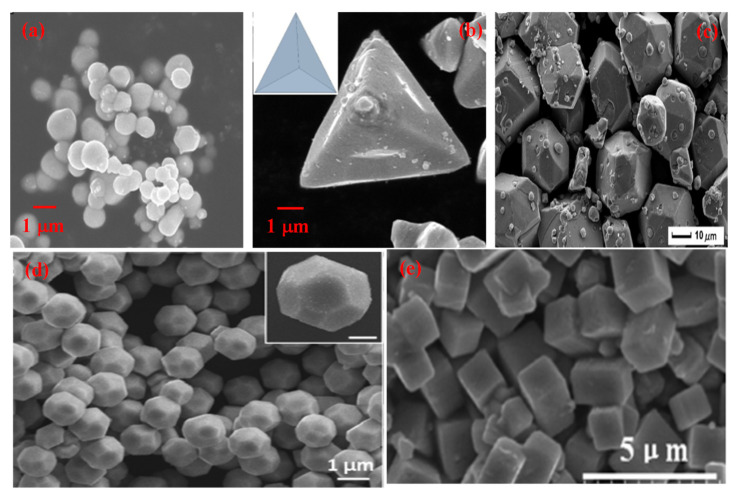
SEM micrographs of different morphologies of Ag_3_PO_4_ (**a**) Sphere (Reprinted with permission from [141]. Copyright 2018 Elsevier), (**b**) Tetrahedral (Reprinted with permission from [142]. Copyright 2018 Elsevier), (**c**) Polyhedral (Reprinted with permission from [137] Copyright 2013 American Chemical Society), (**d**) Dodecahedral (Reprinted with permission from [143]. Copyright 2017 American Chemical Society), (**e**) Cubic (Reprinted with permission from [144]. Copyright 2015 Elsevier).

**Table 4 molecules-27-07778-t004:** Details of different morphologies of nanostructured Ag_3_PO_4_.

S. NO.	Material & Morphology	Method	Application	Performance	Reference
		Spherical			
1	Ag_3_PO_4_/Spherical	Continuous flow synthesis	Microfluidic photocatalytic dye-degradation, microreactor under visible light illumination	97% within 15 min	[145]
2	Ag_3_PO_4_/Spherical	Precipitation method	Phenol, BSP, visible light, 400-W metal halide lamp	82% phenol within 12 min, 81% Bisphenol within 10 min	[146]
3	Ag_3_PO_4_/Spherical	Precipitation	Rh.B. Xenon lamp (15 W), visible light	88% within 35 min,	[147]
4	Ag_3_PO_4_/Spherical	Sol-gel	phenol under visible light irradiation with a 1000 W Xenon lamp	42% within 60 min,	[142]
5	Ag_3_PO_4_/Spherical	Precipitation method at room temperature, 500 W Xenon lamp	MO under visible light irradiation	35%, within 15 min	[141]
6	Ag_3_PO_4_/Spherical	Precipitation method	MB, 5 W compact fluorescent lamp, visible light	78%, within 70 min	[13]
7	Ag_3_PO_4_/Spherical	Sol-gel	6 W/649 fluorescent lamp	Sulfamethoxazole, 100% after 15 min	[148]
8	Ag_3_PO_4_/Spherical	Coprecipitation method at 20 °C	RhB, visible light, 300 W Xenon lamp	66% within 6 min	[149]
9	Ag_3_PO_4_/Spherical	Precipitation method	Congo red (CR) under visible light irradiation, 400 W metal halogen lamp	96%, within 210 min	[150]
10	Ag_3_PO_4_/Spherical	Precipitation method	CR, visible light irradiation with a 350 W Xenon lamp	91%, within 14 min	[151]
11	Ag_3_PO_4_/Spherical	Precipitation method	RhB and MO dyes in 10 mgL^−1^, WLED with a luminous flux (Φv) of 85 l m	100% within 30 min	[152]
		Tetrahedral			
12	Ag_3_PO_4_/Tetrahedral	Sol-gel	phenol under visible light irradiation with a 1000 W Xenon lamp	70% within 60 min,	[142]
13	Ag_3_PO_4_/Tetrahedral	Sol-gel method	MO, visible light, 500 W Xenon lamp	100% within 90 min	[143]
14	Ag_3_PO_4_/Tetrahedral	Precipitation method	MB, MO, RhB Visible light, 500 W Xenon lamp	100% MB, 93% MO, 100% RhB within 6 min	[153]
15	Ag_3_PO_4_/Tetrahedral	Precipitation method	MB, under visible light irradiation, 300 W Xenon lamp	88% within 12 min	[154]
16	Ag_3_PO_4_/Tetrahedral	Ion exchange in the ethanol-water solvent at room temperature	RhB, visible-light provided by a 250 W Xenon lamp	100% within 24 min	[155]
		Dodecahedral			
17	Ag_3_PO_4_/dodecahedral	Sol-gel	phenol under visible light irradiation with a 1000 W Xenon lamp	100% within 60 min,	[142]
18	Ag_3_PO_4_/dodecahedral	Precipitation method	CR under visible light irradiation, 400 W metal halogen lamp	85%, within 210 min	[150]
19	Ag_3_PO_4_/dodecahedral	Precipitation method	MB), RhB, and reactive orange (RO), visible light, TL-D/35 fluorescent tube (18 W, Philips)	90% MB, 82% RhB, 22% RO within 60 min	[156]
20	Ag_3_PO_4_/dodecahedral	Sol-gel method	MO, visible light, 500 W Xenon lamp	78% within 90 min	[143]
21	Ag_3_PO_4_/dodecahedral	Hydrothermally processed at 150 °C for 24 h	RhB, UV illumination, 15 W UV germicidal irradiation lamps	99.55% within 120 min	[157]
22	Ag_3_PO_4_/dodecahedral	Precipitation method	MB, MO, RhB Visible light, 500 W Xenon lamp	93% MB, 62% MO, 100% RhB within 18 min	[153]
		Polyhedral			
23	Ag_3_PO_4_/Polyhedral	Hydrothermal at 120 °C for 12 h	RhB, visible light, 300 W Xenon lamp	97.83% in 6 min	[149]
24	Ag_3_PO_4_/Polyhedral	Sol-gel	RhB, visible light, 350 W Xenon lamp	100% in 4 min	[137]
25	Ag_3_PO_4_/Polyhedral	Precipitation method	MO, 300 W halogen lamp	85% of MO within 15 min	[158]
26	Ag_3_PO_4_/Polyhedral	Sol-gel	phenol under visible light irradiation with a 1000 W Xenon lamp	100% within 60 min	[142]
27	Ag_3_PO_4_/Polyhedral	Conventional ion exchange/precipitation method	phenol under visible light irradiation with a 35 W Xenon lamp	100% within 120 min	[159]
		Cubic			
28	Ag_3_PO_4_/Cubic	Ion exchange method	RhB, Sunlight	100% within 10 min	[144]
29	Ag_3_PO_4_/Cubic	Sol-gel method	MO, visible light, 500 W Xenon lamp	65% within 90 min	[143]
30	Ag_3_PO_4_/Cubic	Precipitation method	crystal violet (CV) and MB, MO and orange G (OG) with visible irradiation, 125 W high-pressure HG lamp	93.0% of CV, within 30 min, 98% MB, MO 79.4%, OG 57.3% within 50 min	[160]
31	Ag_3_PO_4_/Cubic	Precipitation method	MB, MO, RhB Visible light, 500 W Xenon lamp	91% MB, 32% MO, 78% RhB within 18 min	[153]
32	Ag_3_PO_4_/Cubic	Simple ion-exchange deposition method	RhB, Visible light, 300 W Xenon	92% within 30 min	[161]
33	Ag_3_PO_4_/Cubic	Hydrothermal treatment at 160 °C for 3 h	MB and RhB, sunny light between 10 am to 2 pm in the summer	81% within 90 min	[162]

Variation in the synthesis methodology acts as an important aspect of the development of a particular morphology. Different morphologies of silver phosphate have been synthesized by various approaches, such as coprecipitation, sol-gel, hydrothermal, and ion exchange methods (Table 4). Nanospheres of silver phosphate were prepared mostly using the coprecipitation method [13,133,141,147,149,152,163]; however, the sol-gel route was also employed [142]. In less than 1 h, the complete mineralization was achieved using silver phosphate nanospheres catalytically. Tetrahedral morphology of silver phosphate was obtained using sol-gel method [143], coprecipitation [153] and ion exchange method [155]. The removal of aromatic dye contaminant was accomplished within an hour using tetrahedral nanocrystals. Dodecahedral morphology of silver phosphate was synthesized using sol-gel [142,143], coprecipitation [153] and hydrothermal route [157]. It takes about two hours for the complete mineralization of organic pollutants from an aqueous medium photocatalytically. Polyhedral morphology was obtained using the hydrothermal [149], sol-gel [142] and coprecipitation methods [158]. It takes less than an hour for the degradation of organic contaminants from an aqueous medium by employing the polyhedral morphology of silver phosphate. It has been reported [137] that Ag_3_PO_4_ with different morphology shows entirely different exposed facets. Hence it is important to compare the efficiency of photocatalysis by different morphologies such as spheres, tetrahedral, dodecahedral, polyhedral and cubic morphologies of Ag_3_PO_4_ (Figure 8).

Figure 8 denotes a comparative account of different morphologies of silver phosphate. It can be seen that polyhedral morphology takes five hours for synthesis, whereas nanospheres, tetrahedral, and dodecahedral morphologies of silver phosphate can be prepared within an hour. The temperature of synthesis required by cubic morphology was reported to be the highest (160 °C), while other morphologies show moderate values. The photocatalytic performance of all the reported morphologies of silver phosphate was good. In the case of time required for photocatalytic degradation of organic compounds, dodecahedral and nanospheres shows the highest value, whereas tetrahedral, polyhedral and cubic morphologies denote moderate value. Surface area values of polyhedral morphology are the highest among the reported, while dodecahedral and nanospheres had the lowest values among reported.

Our analysis indicates that polyhedral morphology exhibits the highest activity compared to any other morphology. The photocatalytic activity follows the order of polyhedral > tetrahedral > cubic > dodecahedral and sphere. It can be seen from Figure 8 that polyhedral morphology shows 100% activity within 4 min and probably due to its higher surface area (18 m^2^/g). The polyhedral morphology was obtained using a simple eco-friendly sol-gel method at room temperature [142]. Other morphologies also show good photocatalytic activity, but the required time is high, and this may be due to low surface area and poor accessibility of reactants to reach the active sites.

### 2.5. Zinc Titanate (ZnTiO_3_) Nanostructures

**Figure 9 molecules-27-07778-f009:**
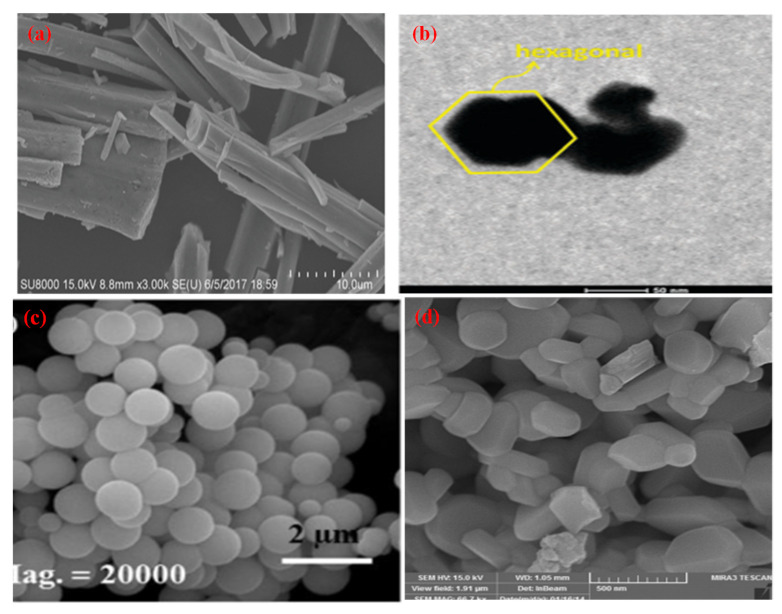
SEM & TEM micrographs of different morphologies of ZnTiO_3_ (**a**) Nanorod (Reprinted with permission from [164]. Copyright 2018 Royal Society of Chemistry), (**b**) Hexagonal (Reprinted with permission from [165]. Copyright 2014 Royal Society of Chemistry), (**c**) Nanosphere (Reprinted with permission from [166]. Copyright 2016 Wiley), (**d**) Particle (Reprinted with permission from [167]. Copyright 2016 Elsevier).

Metal titanates-based dielectric materials with the general formula ATiO_3_ (A denotes metal) are known for different values of permittivity ranging from 7 to more than 5000 under temperature variations. ZnTiO_3_ system (hexagonal ilmenite) has been studied earlier mainly due to its striking features for practical applications in microwave resonators, mobile phones, and satellite communication systems. However, the above studies need well-sintered materials with large particle sizes. Various morphologies of ZnTiO_3_ NMs have been developed (Figure 9) due to their special features for their application as photocatalysts with a lower band gap (3.08 eV) than TiO_2_ (3.2 eV). Three compounds are known in the ZnO-TiO_2_ system such as Zn_2_TiO_4_ (cubic spinel), ZnTiO_3_ (hexagonal ilmenite) and Zn_2_Ti_3_O_8_ (cubic defect spinel). The thermodynamically stable phase for ZnTiO_3_ at low temperatures is a cubic phase which changes to a hexagonal phase at high temperatures (800 °C) and also decomposes into Zn_2_TiO_4_ and TiO_2_ at a temperature of 900 °C. ZnTiO_3_ decomposes at high temperatures since Zn volatilizes and creates non-stoichiometry. Thus, it is challenging to synthesize pure ZnTiO_3_.

ZnTiO_3_ has two types of crystal structures, i.e., cubic and hexagonal, out of which this hexagonal structure shows the best photocatalytic performance. The pure form of hexagonal ZnTiO_3_ can be obtained by using high-temperature solid-state reactions as well as low-temperature methods such as sol-gel, hydrothermal, modified alcoholysis, carbothermal and molten salt methods. Hexagonal ZnTiO_3_ is a UV-active photocatalyst with a band gap of 3.08 eV. The hydrothermal method was used in the synthesis of zinc titanate [168]. Alternately pure zinc titanate can also be prepared by the conventional solid-state method at high temperatures. Other methods, such as mechanochemical activation, molten salt synthesis, and microbead milling, were also applied in the synthesis of ZnTiO_3_ [169], but they resulted in irregular size, morphology, and grain size. Sol-gel technique has been preferred mostly by researchers in the synthesis of zinc titanate due to its low cost, high yield, small processing time, homogeneity and high purity of the end product. It can be useful in making thin films, membranes and multicomponent oxide materials. Generally, metal alkoxides are used as precursors in the sol-gel process. Further, the metal alkoxide is subjected to hydrolysis and polycondensation to obtain the desired combinations. The resultant material obtained by the sol-gel process has a large specific surface area and pore volume, which is the essential requirement of the material to be used as a catalyst.

**Table 5 molecules-27-07778-t005:** Details of different morphologies of nanostructured ZnTiO_3_.

S. NO.	Material & Morphology	Method	Application	Performance	Reference
		Particles			
1	ZnTiO_3_/Particle	Sol-gel	Phenol, under a 100 W incandescent visible lamp	100% in 300 min	[170]
2	ZnTiO_3_/Particle	Hydrothermal, at 180 °C for 8 h	MB, under a 350 W Xenon visible lamp	29.7% in 120 min	[168]
3	ZnTiO_3_/Particle	Sol-gel	4-chlorophenol, under natural sunlight	67% in 45 min	[171]
4	ZnTiO_3_/Particle	Sonochemical method	Rh), under a 70 W LED visible lamp	36% in 150 min	[172]
5	ZnTiO_3_/Particle	Sol-gel	MO, under a 400 W UV lamp	70% in 60 min	[167]
6	ZnTiO_3_/Particle	Solvothermal, at 180 °C 24 h	RhB & MO, under a 400 W halide visible lamp	17% of Rh B, 3% of MO in 90 min	[157]
		Rod			
7	ZnTiO_3_/Rod	Microwave	RhB, under a 150 W Xenon visible lamp	93% in 180 min	[173]
8	ZnTiO_3_/Rod	Sol-gel	RhB, under a 50 W high-pressure Hg lamp	97% in 70 min	[174]
9	ZnTiO_3_/Rod	Precipitation method	RhB, under sunlight	71% in 60 min	[164]
10	ZnTiO_3_/Rod	Hydrothermal, 120 °C for 24 h	MO, under a 500 W Xenon lamp	99.3% in 20 min	[175]
11	ZnTiO_3_/Rod	Sol-gel	RhB & crystal violet, under sunlight	98% of CV in 60 min & 77% of RhB in 90 min	[166]
		Spherical			
12	ZnTiO_3_/Spherical	Sol-gel	Methyl violet, under sunlight	97% in 120 min	[169]
13	ZnTiO_3_/Spherical	Sol-gel	H_2_ production, under a 125 W high-pressure Hg UV lamp	110 µmol/h, 60% in 3600 min	[176]
14	ZnTiO_3_/Spherical	Sol-gel	MB, under a 150 W UV lamp	33% in 3600 min	[177]
15	ZnTiO_3_/Spherical	Sol-gel	Norfloxacin (NOR) and MO, under a 300 W Xenon visible lamp	95% of NOR & 46% of MB in 60 min	[178]
16	ZnTiO_3_/Spherical	Sol-gel	MB, under sunlight	76% in 60 min	[179]
		Hexagonal			
17	ZnTiO_3_/Hexagonal	Sol-gel	p-nitrophenol, under sonocatalytic activity	74.8% in 180 min	[180]
18	ZnTiO_3_/Hexagonal	Sol-gel	RhB, under sunlight	35% in 180 min	[165]

Varying morphologies of zinc titanates were synthesized using several low-temperature methods. It can be observed from Table 5 that the method of synthesis plays an essential role in the formation of different morphologies such as nanorods, nanoparticles, nanospheres and hexagonal nanostructures. Zinc titanate nanoparticles were synthesized by using a sol-gel approach [167,170,171], hydrothermal/solvothermal process [168,181] sonochemical method [172]. The temperature of synthesis of nanoparticles was reported as 180 °C in general for the hydrothermal method, whereas sol-gel and sonochemical methods require a lower temperature. Zinc titanate nanoparticles normally take an hour to degrade 70% of the organic pollutants. The nanorods were obtained using sol-gel [166,174], microwave technique [173], coprecipitation method [164], and the hydrothermal process [173]. The hydrothermal process consumes more energy as compared to sol-gel and coprecipitation techniques. The photocatalytic performance shows that the nanorod structure of zinc titanate requires 20 min [182] for the complete mineralization of organic pollutants. Zinc titanate nanospheres have been synthesized using the sol-gel technique in general. This morphology, as reported by Kong et al. [169], takes 2 h for the complete mineralization of aromatic compounds photocatalytically. In the case of hexagonal nanostructures synthesis by the sol-gel technique, [165,180] is mostly preferred. As per the available report, the time taken for the degradation of aromatic compounds is more than three hours by zinc titanate hexagonal nanostructures. Thus, considering the ease of synthesis by the sol-gel method and the efficiency of the photocatalytic performance of zinc titanate nanorods, it should be the most preferred morphology.

To analyze the different reported morphologies of zinc titanate and to decide the optimal one which is the best for the photocatalytic application, we have chosen nanorods, hexagonal nanostructures, nanospheres and nanoparticles for our assessment (Figure 10). The standard parameters such as surface area, time of synthesis, photocatalytic and catalytic activity time and synthesis temperature have been considered for the assessment.

Figure 10 reveals an assessment of various morphologies of zinc titanate. The nanorod morphology requires maximum time for synthesis (20 h), while other morphologies such as nanospheres, hexagonal and nanoparticles can be prepared in an average of five hours. Hexagonal morphology needs the highest temperature for synthesis (350 °C), whereas nanorods need the lowest values, and nanospheres show the highest values, whereas hexagonal and nanoparticles show a 30% lesser value, comparatively. The time taken for photocatalytic degradation is highest (3 h) for hexagonal morphology; however other morphologies need an hour for complete degradation. The value of the surface area is the highest for nanorod morphology, while nanoparticles and nanospheres show moderate value, and hexagonal morphology denotes a lower surface area, comparatively.

It can be seen from Figure 10 that the nanorod [182] morphology possesses the highest surface area (74 m^2^/g), whereas hexagonal morphology has the lowest surface area (10 m^2^/g) amongst the reported structures [180]. Complete mineralization of the organic pollutants was achieved photo-catalytically within 20 min in the case of nanorod morphology, while various other morphologies require more than an hour to achieve the same target. The time of synthesis of the nanorod morphology was 24 h which is the highest amongst the other reported morphologies, whereas nanospheres can be prepared within one hour by using a similar hydrothermal technique. It is the lowest reported time of synthesis compared to other morphologies mentioned. Thus, by taking into account the variation in the assessment parameter reported, it can be seen that nanorod morphology shows the best result and can be considered the best photocatalytic material in the case of zinc titanate. The order of preference based on efficiencies is nanorods [182], nanospheres [178], nanoparticles [167] and lastly, hexagonal nanostructures [180] of zinc titanate.

Figure 11 shows the comparative analysis of photocatalytic performance and mineralization time of TiO_2_, ZnO, CdS, Ag_3_PO_4_ and ZnTiO_3_ nanomaterials at a glance. It can be seen that silver phosphate shows the best photocatalytic activity amongst the studied nanomaterials based on the minimum time taken (less than ten minutes) for the complete mineralization of organic compounds. Though the band gap of silver phosphate (2.28 eV) and cadmium sulphide (2.3 eV) are similar, their performances are poles apart, and the reason may be the morphology formed by these nanomaterials. Further, based on an overall comparison with the other photocatalyst, the order of preference can be assigned as silver phosphate (Ag_3_PO_4_), zinc oxide (ZnO), zinc titanate (ZnTiO_3_), titanium dioxide (TiO_2_) and cadmium sulphide (CdS).

**Figure 11 molecules-27-07778-f011:**
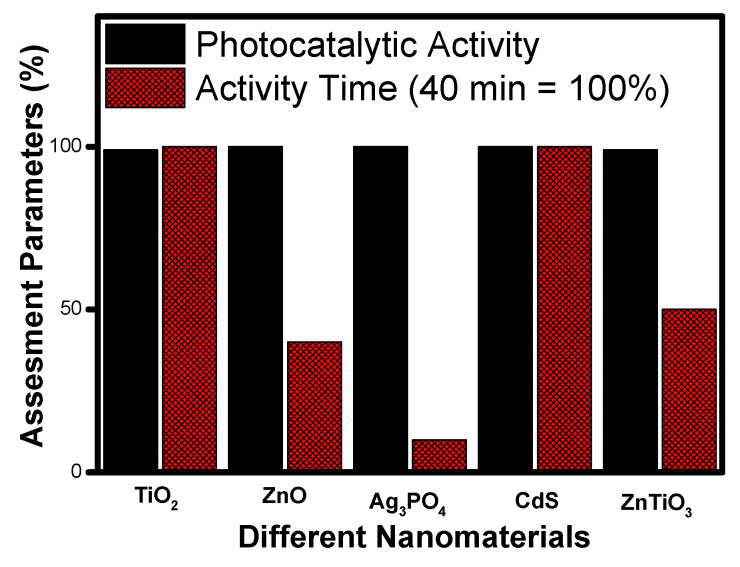
Comparative photocatalytic activity among different photocatalysts based on degradation time.

**Table 6 molecules-27-07778-t006:** Mechanism of photocatalysis for TiO_2_, ZnO, Ag_3_PO_4_, CdS and ZnTiO_3_ photocatalysts nanomaterials photocatalysts.

S. No.	Materials	Photocatalysis Mechanism	Ref.
1.	TiO_2_	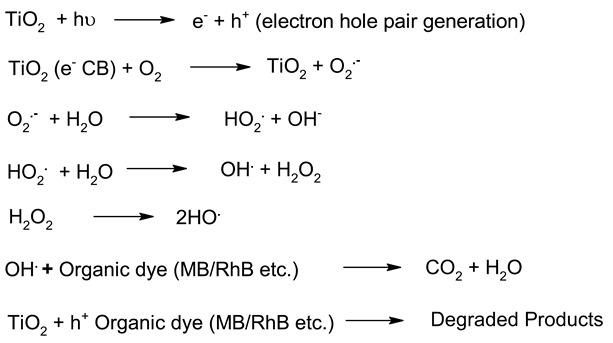	[183]
2.	ZnO	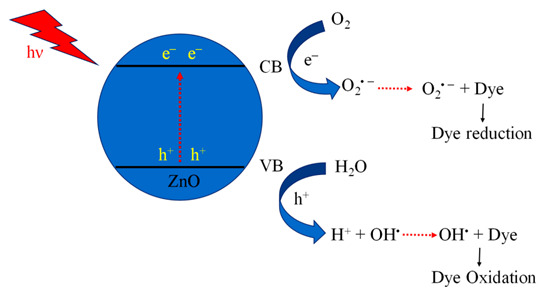	[184]
3.	CdS	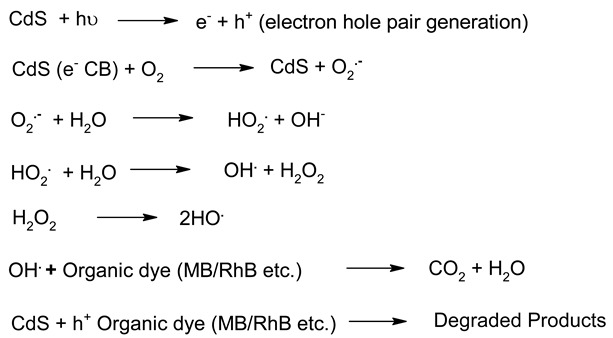	[185]
4.	Ag_3_PO_4_	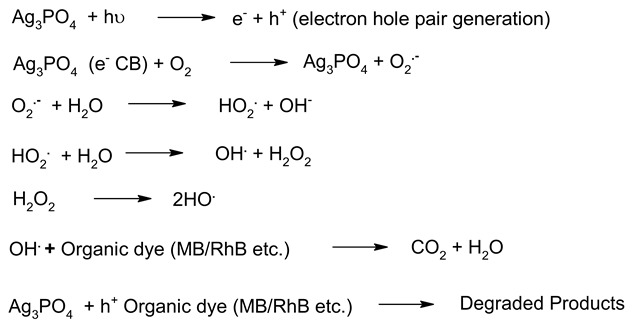	[186]
5.	ZnTiO_3_	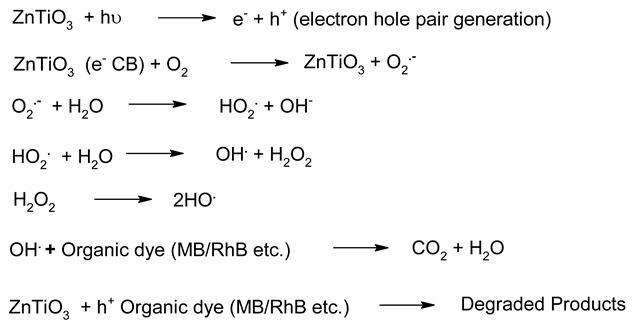	[187]

**Table 7 molecules-27-07778-t007:** Important characteristics of photocatalyst TiO_2_, ZnO, Ag_3_PO_4_, CdS and ZnTiO_3_.

S. No.	Photocatalytic Materials	Band Gap Energy (E_g_) eV	Photocatalytic Performances	Ref.
1.	TiO_2_	E_g_ ~ 3.1	The synthesised TiO_2_ nanotubes possess better photocatalytic activity than the as-prepared counterparts because of the larger surface area and good crystallinity. The emission stability of the catalyst also validates that TiO_2_ nanotubes could find potential applications in cold-cathode-based electronics. Literature shows that the decoration of TiO_2_ nanotubes by noble metal nanoparticles (such as Au, Ag, and Pt) also enhances its photocatalytic activity.	[45,188,189]
2.	ZnO	E_g_ ~ 3.35	Qu et al. Synthesised various shapes and morphology of ZnO nanomaterials using different ultrasonic processes. The study revealed that ZnO nanoflower morphology shows excellent photocatalytic activity.	[97,190]
3.	Ag_3_PO_4_	E_g_ ~ 2.45	Morales et al. synthesised silver phosphate microcrystals with polyhedral morphologies and its higher surface area played an important role in higher photocatalytic activity.Geng et al. successfully prepared the polyhedral morphology of the Ag_3_PO_4_ microcrystal structure. The photocatalytic activity study confirmed its excellent photocatalytic ability.	[191,192]
4.	CdS	E_g_ ~ 2.4	Ganesh et al. synthesised CdS nanoflowers. The nano flowers of the CdS materials showed better photocatalytic activity in visible light. Wang et al. recently studied Ti_3_C_2_ MXene@CdS based nanoflowers composites heterostructures. It shows steady photoluminescence intensity and a longer fluorescence lifetime.	[100,193]
5.	ZnTiO_3_	E_g_ ~ 2.9 (indirect) 3.59 (direct)	Dutta et al. reported the Ag-doped ZnTiO_3_ nanorods. The photocatalytic results show improved photocatalytic activity. Chuaicham et al. recently studied ZnTiO_3_ mixed metal oxide. The higher photocatalytic activity was observed for phenol photodegradation.	[173,194]

**Table 8 molecules-27-07778-t008:** Photocatalytic Stability and biocompatibility of different NP photocatalysts.

Type of NPs	Stability and Recyclability	Biocompatibility	Ref.
TiO_2_	The photostability of TiO_2_ for phenol degradation in four cycles remained constant.	Biocompatible, supports osteoblast-like cell formation, and can be used in biomedical applications.	[195,196,197]
ZnO	RhB aqueous solution for five cycles, good recyclability.	high bactericidal efficacy along with good cytocompatibility	[64,76,98,198]
Ag_3_PO_4_	Photocatalytic efficiency remained consistently high after four cycles [11].	Spectacular biocompatibility and good immunosensor sensitivity, low toxicity.	[137,199]
CdS	The catalytic activity of the photocatalyst remained constant.	Environment-friendly and economical photocatalyst	[200,201]
ZnTiO_3_	The efficacy of the photocatalyst remained constant.	Excellent antitumor ability and good biocompatibility	[202,203,204]

### 2.6. Challenges during the Application of NPs for Photochemical Reactions

Irrespective of the worldwide interest in the use of NPs in photocatalysis for the conversion of visible light energy for chemical conversions, this technology has not been successful in its commercialisation and large-scale applications. The long-term exposure of photocatalytic NPs under constant irradiation may lead to changes in the morphology of the material, which may affect the performance of the photocatalysts. To enhance the efficiency of the photocatalysts, in the majority of cases, the photocatalysis process requires a constant light source, which can be enabled through artificial setups. It has been observed that the supported NPs perform better than the bare ones during photocatalysis [205]. The challenges in the area of photocatalysis can only be addressed through research and a sustainable approach. There is a need for an effective implementation strategy to be worked for MNPs’ visible light photocatalysis. The use of scavengers in photocatalysis may help in improving the efficiency of the photocatalysts. Effective protocols are required for the large-scale synthesis of MNPs, and their further disposal after use may encourage sustainable practices in this area of work. The noteworthy research on greener processes for the synthesis of MNPs can be a significant step in the large-scale implementation of photocatalysis processes for environmental mitigation [206,207]. The typical mechanistic pathway of the photochemical reactions (Table 6) reveals the role of different nanomaterial photocatalysts during dye degradation. The generation of free radicals by the photochemical effect initiates dye degradation, and it depends on the band gap of the nanomaterial photocatalyst. It can be seen from Table 7 that the recyclability and band gap of the different nanomaterial photocatalysts should be considered for the selection of the catalyst for photochemical reactions. Reusability, photochemical stability and biocompatibility (Table 8) are the essential characteristics of sustainable photochemical catalysts. 

In photocatalysis, the efficacy of the catalyst depends on its ability to lower the activation energy by offering an efficient reaction pathway. The metal oxide-based photocatalysts work on these characteristics for the degradation of organic contaminants [208]. The environmentally benign photocatalyst materials with low cost and availability of other chemicals such as solvents, dopants etc., and their environmental impact is a challenging task. But the phenomenal growth in this field of work can make available smart and sustainable materials for environmental photocatalysis [209].

### 2.7. Future Direction for the Application of Nanomaterial Photocatalyst

Presently, nanotechnology manipulates and creates matter to fabricate materials on the nanometre scale. Nanomaterials are used in the interdisciplinary field, and they have societal implications for a wide range of scientific and engineering disciplines. In the past few decades, nanotechnology has been applied in various sectors like; environment, materials science, energy, electronics, agriculture, healthcare, biotechnology, and information technology. Certain nanomaterials have the excellent potential to revolutionize wastewater treatment through photocatalysis and other techniques. Because of their remarkable characteristics, such as effective adsorption and photocatalytic properties, nanomaterial oxides are being used commercially for wastewater treatment extensively. For technology related to clean environment, and health care, like drug delivery and cancer therapy, nanotechnology has commercially provided scope for existing firms to upgrade their products and services. Several oxides, sulfides, tantalates, phosphates and niobates are used commercially for multiple cures [210,211]. Silver phosphate and cadmium sulfide are used commercially as photocatalysts in novel photoreactors for better catalytic activity [212].

## 3. Conclusions

Effective morphology of the nanomaterial is an essential requirement in photocatalysis for application in dye degradation or conversion of solar energy into chemical energy. The morphology and size of nanomaterial not only reduce the band gap of the photocatalyst but also enhances its photocatalytic performance in the degradation of complex organic contaminants. The activity of photocatalysts solely relies upon band gap reduction and effective utilisation of photons into photocatalytic conversions when subjected to the required wavelength. Morphology played an important role in the band gap reduction of nanomaterials. Various morphologies of a particular nanomaterial show different photocatalytic performances.

(i)The analysis of literature data indicates that titanium oxide (TiO_2_) nanotube morphology emerged as the best material. It has excellent photocatalytic performance and takes minimum time for the degradation of contaminants. Due to its high surface area, the active centres are easily accessible during photocatalysis;(ii)Similar criteria, when applied to other nanomaterials, show zinc oxide (ZnO) and cadmium sulphide (CdS) nanomaterial having nanoflower morphology as the best among other reported;(iii)Nanorod morphology appeared as the best morphology for zinc titanate (ZnTiO_3_) for photocatalytic applications;(iv)Silver phosphate (Ag_3_PO_4_) shows polyhedral morphology as the best performer on all given counts and appears to be the best morphology for a photocatalyst.

Thus, based on its impact on photocatalytic performance, the morphology of the nanomaterials should be considered an essential aspect in the selection of a photocatalyst. We are positive that the analysis applied in this research article can be useful in the study of other photocatalytic systems and bridge the gap between academia and industry in terms of the selection of photocatalysts for industrial applications.

## Figures and Tables

**Figure 2 molecules-27-07778-f002:**
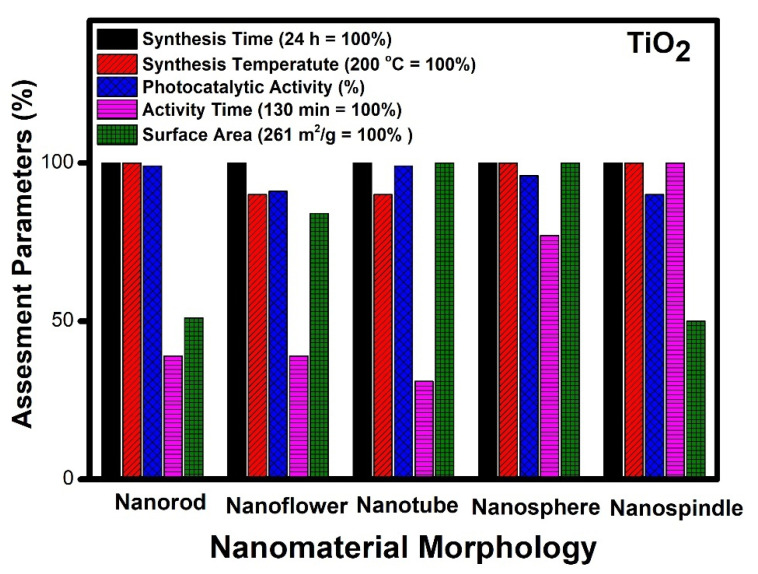
Assessment of different morphologies of TiO_2_ based on various parameters.

**Figure 4 molecules-27-07778-f004:**
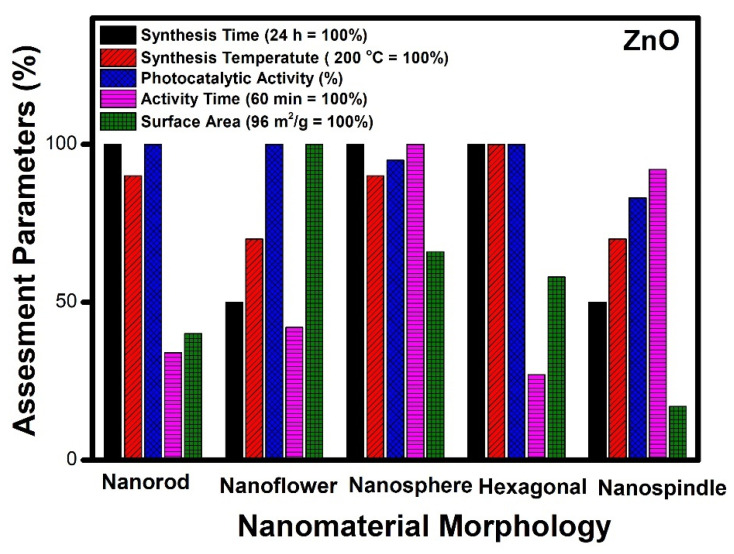
Assessment of different morphologies of ZnO based on various parameters.

**Figure 6 molecules-27-07778-f006:**
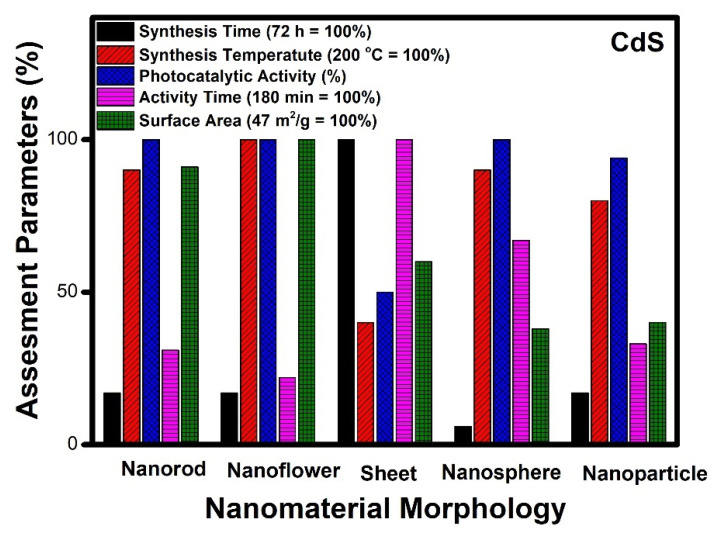
Comparative study of different morphologies of CdS based on various assessment parameters.

**Figure 8 molecules-27-07778-f008:**
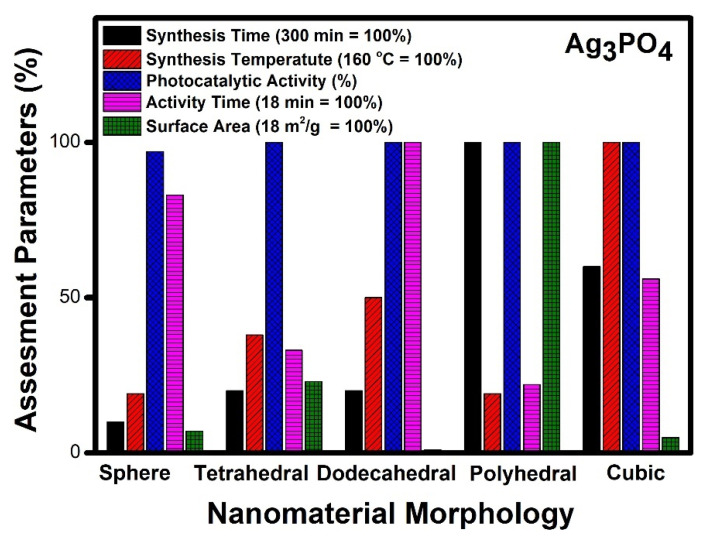
Comparative study of different morphologies of Ag_3_PO_4_ based on various assessment parameters.

**Figure 10 molecules-27-07778-f010:**
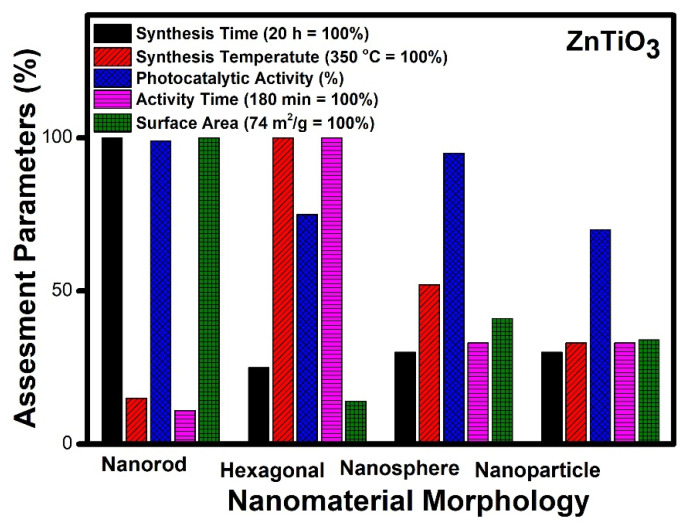
Comparative study of different morphologies of ZnTiO_3_ based on various assessment parameters.

## Data Availability

The study did not report any external dataset link.

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
