# Peer review of "Assessment of Performance of Photocatalytic Nanostructured Materials with Varied Morphology Based on Reaction Conditions"

_molecules, 2022, doi:10.3390/molecules27227778_

Round 1

Reviewer 1 Report

The paper entitled “Assessment of performance of photocatalytic nanostructured materials with varied morphology based on reaction conditions” was prepared by Ganguli et al performance of photocatalytic nanostructured materials with varied morphology based on reaction conditions. The work is interesting, By the way, I recommend a major revision and it is needed to address the following issue:

-subscripts in many parts didn’t use properly, for example, Table 1

-Please address the limitation of using metal NPs in photocatalytic, it suggests adding a section “Challenges”

- please revise the sentences and remove the grammatical errors

- There are many studies investigating the importance of the topic, Please add these references to your introduction and discussion parts of the manuscript and compare and bold your study novelty: https://doi.org/10.1016/j.jece.2021.107091, https://doi.org/10.1021/acs.orglett.9b01230, https://doi.org/10.1039/D1CP03943A, https://doi.org/10.1002/slct.202002978

- what is the suggestion of this study for future works?

- The NPs stability and biocompatibility need to investigate by the authors

- There are some spelling errors and logic problems in the text that need attention. Moreover, the typos in the man

-The conclusions of the work are not well established and are not solid scripts that need to be double-checked. The conclusion that needs to be upgraded may be Including a discussion part on the cost, and possible side effects or limitations in the use of NPs to increase the impact of the paper.

Author Response

Reviewer 1

The paper entitled “Assessment of performance of photocatalytic nanostructured materials with varied morphology based on reaction conditions” was prepared by Ganguli et al performance of photocatalytic nanostructured materials with varied morphology based on reaction conditions. The work is interesting, By the way, I recommend a major revision and it is needed to address the following issue:

  1. subscripts in many parts didn’t use properly, for example, Table 1

Ans: The subscripts and superscripts are now corrected in the entire manuscript. Thank you.

  1. Please address the limitation of using metal NPs in photocatalytic, it suggests adding a section “Challenges”

Ans:

2.6 Challenges during application of NPs for photochemical reactions

Irrespective of the worldwide interest in the use of NPs in photocatalysis for the conversion of visible light energy for chemical conversions, this technology has not been successful for its commercialization and large-scale applications. The long-term exposure of photocatalytic NPs under constant irradiation may lead to changes in the morphology of the material which may affect the performance of the photocatalysts. To enhance the efficiency of the photocatalysts, in majority of the cases, photocatalysis process require constant light source, which can be enabled through artificial setups. It has been observed that the supported NPs perform better than bare ones during photocatalysis [216]. The challenges in the area of photocatalysis can only be addressed through research and sustainable approach. There is need of effective implementation strategy to be worked for MNPs visible light photocatalysis. The use of scavengers in the photocatalysis may help in improving the efficiency of the photocatalysts. The effective protocols are required for the large-scale synthesis of MNPs and its further disposal after use my encourage the sustainable practices in this area of work. The noteworthy research on greener processes for the synthesis of MNPs can be a significant step in the large-scale implementation of photocatalysis processes for environmental mitigation [217][218]. The typical mechanistic pathway of the photochemical reactions (Table 6) reveals the role of different nanomaterial photocatalysts during dye degradation. The generation of free radical by photochemical effect initiates the dye degradation and it depend on the band gap of the nanomaterial photocatalyst. It can be seen from the Table 7 that the recyclability and band gap of the different nanomaterial photocatalyst should be considered for the selection of the catalyst for photochemical reactions.  Reusability, photochemical stability and biocompatibility (Table 8) are the essential characteristics of sustainable photochemical catalysts.        

In photocatalysis the efficacy of the catalyst depends on its ability to lower the activation energy by offering efficient reaction pathway. The metal oxide based photocatalysts work on these characteristics for the degradation of organic contaminants [219]. The environmentally benign photocatalyst materials with low cost and availability of other chemicals such as solvents, dopants etc. and their environmental impact is a challenging task. But the phenomenal growth in this field of work can make available smart and sustainable materials for environmental photocatalysis [220].

  1. please revise the sentences and remove the grammatical errors

Ans: The sentences are rephrased and grammar is corrected.

  1. There are many studies investigating the importance of the topic, please add these references to your introduction and discussion parts of the manuscript and compare and bold your study novelty:

https://doi.org/10.1016/j.jece.2021.107091, https://doi.org/10.1021/acs.orglett.9b01230, https://doi.org/10.1039/D1CP03943A, https://doi.org/10.1002/slct.202002978

Ans: The research articles mentioned are incorporated in the introduction part of the manuscript. Thank you.

[3]        C. Zhao, M. Xi, J. Huo, C. He, Phys. Chem. Chem. Phys. 2021, 23, 23219.

[4]        M.R. Wang, L. Deng, G.C. Liu, L. Wen, J.G. Wang, K. Bin Huang, H.T. Tang, Y.M. Pan, Org. Lett. 2019, 21, 4929.

[5]        S. Nikazar, M. Barani, A. Rahdar, M. Zoghi, G. Kyzas, ChemistrySelect 2020, 5, 12590.

[29]      Y. Wang, X. Wu, J. Liu, Z. Zhai, Z. Yang, J. Xia, S. Deng, X. Qu, H. Zhang, D. Wu, J. Wang, C. Fu, Q. Zhang, J. Environ. Chem. Eng. 2022, 10.

  1. what is the suggestion of this study for future works?

Ans: The speed and enormous availability of the research data on photocatalysts may confuse the user in the selection of particular photo catalyst for the photocatalytic reaction. Herein we have tried to simplify the process of selection of the photocatalyst amongst the available and most popular photocatalysts with proper reasoning. The most popular and well-studied photocatalysts morphology were selected to come to some reasonable conclusion based on the catalyst structure, reaction parameters of these catalysts. The analogy applied can be effective in the study of different morphologies for the different photocatalysts also. The study can be effective in the selection of best nanostructured materials amongst several materials of similar composition reported here based on the critical review of catalytic characteristics for the photochemical reactions. Further it also indicates that structural arrangement, size and geometrical shape have great impact on reactivity of the photocatalytic materials. 

  1. The NPs stability and biocompatibility need to investigate by the authors

Ans: Table 8  Photocatalytic Stability and biocompatibility of different NP photocatalysts.

Type of NPs

Stability and Recyclability

Biocompatibility

Ref.

TiO2

The photostability of TiO2 for phenol degradation in four cycles remained constant.

Biocompatible, supports osteoblast like cells formation, and can be used in biomedical applications.

[45][205]–[207]

ZnO

RhB aqueous solution for 5 cycles, good recyclability.

high bactericidal efficacy along with good cyto-compatibility

[86][97][208][209]

Ag3PO4

Photocatalytic efficiency remained consistently high after four cycles (ref 11).

Spectacular biocompatibility and good immunosensor sensitivity, low toxicity.

 [152][210]

CdS

Catalytic activity of the photocatalyst remained constant.

Environment-friendly and economical photocatalyst

[211][212]

ZnTiO3

Efficacy of the photocatalyst remained constant.

Excellent antitumor ability and good biocompatibility

[213]–[215]

  1. There are some spelling errors and logic problems in the text that need attention. Moreover, the typos in the man

Ans: The typos and grammatical errors are corrected. Thank you.

  1. The conclusions of the work are not well established and are not solid scripts that need to be double-checked. The conclusion that needs to be upgraded may be Including a discussion part on the cost, and possible side effects or limitations in the use of NPs to increase the impact of the paper.

Ans: The conclusions part is now improved and the content on challenges in the use of Photocatalyst NPs included in the manuscript.

  1. Conclusions

Effective morphology of the nanomaterial is an essential requirement in photocatalysis for application in dye degradation or conversion of solar energy into chemical energy. The morphology and size of nanomaterial not only reduce the band gap of the photocatalyst but also enhances its photocatalytic performance in the degradation of complex organic contaminants. The activity of photocatalysts solely relies upon band gap reduction and effective utilization of photons into photocatalytic conversions when subjected to the required wavelength. Morphology played important role in band gap reduction of nanomaterials. Various morphologies of a particular nanomaterial show different photocatalytic performance.

  1. i) The analysis of literature data indicates that titanium oxide (TiO2) nanotube morphology emerged as the best material. It has excellent photocatalytic performance and takes minimum time for the degradation of contaminants. Due to its high surface area, the active centres are easily accessible during photocatalysis.
  2. ii) Similar criteria when applied to other nanomaterials show zinc oxide (ZnO) and cadmium sulphide (CdS) nanomaterial having nanoflower morphology as the best among other reported.

iii) Nanorod morphology appeared as the best morphology for zinc titanate (ZnTiO3) for photocatalytic applications.

  1. iv) Whereas, silver phosphate (Ag3PO4) shows polyhedral morphology as the best performer on all given counts and appears to be the best morphology for a photocatalyst.

Thus, based on its impact on photocatalytic performance, the morphology of the nanomaterials should be considered an essential aspect, in the selection of a photocatalyst. We are positive that the analysis applied in this research article can be useful in the study of other photocatalytic systems and bridge the gap between academia and industry in terms of the selection of photocatalysts for industrial applications.

Author Response

Reviewer 2

The authors submitted a manuscript of a review about the different morphologies of specific nanomaterials such as titanium dioxide, zinc oxide, silver phosphate, cadmium sulphide and zinc titanate. The different synthesis strategies adopted for a specific morphology have been compared with the photocatalytic performance. The selective examples of the widely accepted photocatalysts such as TiO2, ZnO, Ag3PO4, CdS and ZnTiO3 have been compared based on their photocatalytic performance, the time required for synthesis and reaction temperature. The review is very useful to obtain a broad knowledge of the effective morphology of the nanomaterial as an essential aspect, in the selection of a photocatalyst for application in dye degradation or conversion of solar energy into chemical energy. This review has clear ideas, detailed content, a lot of work, and could be considered for publication. However, the authors should revise their manuscript before acceptance for publication according to the following comments:

Responses to comments

1-As an observation, there are many bibliographic references, but only until 2019.

Ans: The references are now updated by adding new and latest ones. Thank you.

[3]       C. Zhao, M. Xi, J. Huo, C. He, Phys. Chem. Chem. Phys. 2021, 23, 23219.

[4]       M.R. Wang, L. Deng, G.C. Liu, L. Wen, J.G. Wang, K. Bin Huang, H.T. Tang, Y.M. Pan, Org. Lett. 2019, 21, 4929.

[5]       S. Nikazar, M. Barani, A. Rahdar, M. Zoghi, G. Kyzas, ChemistrySelect 2020, 5, 12590.

[190]   C. Yu, F. Chen, D. Zeng, Y. Xie, W. Zhou, Z. Liu, L. Wei, K. Yang, D. Li, Nanoscale 2019.

[192]   G.M. L., Metals (Basel). 2020, 10, 820.

[193]   S. Gautam, H. Agrawal, M. Thakur, A. Akbari, H. Sharda, R. Kaur, M. Amini, J. Environ. Chem. Eng. 2020, 8.

[196]   A. Yumashev, B. Åšlusarczyk, S. Kondrashev, A. Mikhaylov, Energies 2020, 13.

2-Some details should be noted:

Ans: Thank you so much for through reading, appreciating our work and suggesting in detail regarding English usage and typos errors. All the typos’ errors and English usages are now taken care of.

Reviewer 3 Report

In this manuscript, the authors reported that an assessment performance of photocatalytic nanostructured  materials with varied morphology based on reaction conditions. In the present review study, the authors considered both the aspects like morphology with basic considerations and analysed them in detail. Different morphologies of specific nanomaterials such as titanium dioxide, zinc oxide, silver phosphate, cadmium sulphide and zinc titanate have been discussed to bring home the point. Morphologies such as nanorods, nanoflower, nanospindles, nanosheets, nanospheres and nanoparticles were compared within and outside the domain of given nanomaterials. The different synthesis strategies adopted for a specific morphology have been compared with the photocatalytic performance.  Although the approach is interesting, the present paper contains several weak points, originating from the fact that the lack of novelty and poor broader impact (e.g., readership). In fact, the authors should emphasize on their review  novelty compared to other’s work for their assessment performance of photocatalytic nanostructured  materials with varied morphology. There are many similar review papers have been well-discussed about the similar topic, especially titanium dioxide, zinc oxide, silver phosphate, cadmium sulphide and zinc titanate photocatalysts. The list of reference need to be updated especially for those research findings since 2019 – 2022. The recent literature review on modified titanium dioxide, zinc oxide, silver phosphate, cadmium sulphide and zinc titanate with responsive band gap energy and their photocatalysis performance should be summarized in a table as benchmarking purpose and discussed in detail with your review summary findings. In my opinion, technical significance and novelty of the work is still lacking especially on the material such as titanium dioxide, zinc oxide, silver phosphate, cadmium sulphide and zinc titanate.  Authors need to study and review more on this kind of reported work. Furthermore, the details studies on future direction and commercialization perspectives for this potential photocatalysis system with different morphologies of specific nanomaterials in photocatalysis related applications should be determined and reported. The photocatalysis mechanism of different morphologies of specific nanomaterials should be described and discussed accordingly. Furthermore, carefully English correction is necessary for the whole manuscript. The list of reference needs to be updated. In my opinion, the manuscript contains few experimental results is at a moderate scientific level, rendering this in its present form inappropriate for publication in Molecules.

Author Response

Reviewer 3

Reviewer comments

In this manuscript, the authors reported that an assessment performance of photocatalytic nanostructured materials with varied morphology based on reaction conditions. In the present review study, the authors considered both the aspects like morphology with basic considerations and analysed them in detail. Different morphologies of specific nanomaterials such as titanium dioxide, zinc oxide, silver phosphate, cadmium sulphide and zinc titanate have been discussed to bring home the point. Morphologies such as nanorods, nanoflower, nanospindles, nanosheets, nanospheres and nanoparticles were compared within and outside the domain of given nanomaterials. The different synthesis strategies adopted for a specific morphology have been compared with the photocatalytic performance.  Although the approach is interesting, the present paper contains several weak points, originating from the fact that the lack of novelty and poor broader impact (e.g., readership). In fact, the authors should emphasize on their review novelty compared to other’s work for their assessment performance of photocatalytic nanostructured materials with varied morphology. There are many similar review papers have been well-discussed about the similar topic, especially titanium dioxide, zinc oxide, silver phosphate, cadmium sulphide and zinc titanate photocatalysts. The list of reference needs to be updated especially for those research findings since 2019 – 2022. The recent literature review on modified titanium dioxide, zinc oxide, silver phosphate, cadmium sulphide and zinc titanate with responsive band gap energy and their photocatalysis performance should be summarized in a table as benchmarking purpose and discussed in detail with your review summary findings. In my opinion, technical significance and novelty of the work is still lacking especially on the material such as titanium dioxide, zinc oxide, silver phosphate, cadmium sulphide and zinc titanate.  Authors need to study and review more on this kind of reported work. Furthermore, the details studies on future direction and commercialization perspectives for this potential photocatalysis system with different morphologies of specific nanomaterials in photocatalysis related applications should be determined and reported. The photocatalysis mechanism of different morphologies of specific nanomaterials should be described and discussed accordingly. Furthermore, carefully English correction is necessary for the whole manuscript. The list of reference needs to be updated. In my opinion, the manuscript contains few experimental results is at a moderate scientific level, rendering this in its present form inappropriate for publication in Molecules.

Responses to the reviewer comments

  1. In fact, the authors should emphasize on their review novelty compared to other’s work for their assessment performance of photocatalytic nanostructured materials with varied morphology. There are many similar review papers have been well-discussed about the similar topic, especially titanium dioxide, zinc oxide, silver phosphate, cadmium sulphide and zinc titanate photocatalysts.

Ans: Many a times the morphology of the materials is overlooked by the researchers and industry persons in the selection of the best catalyst for the photochemical reactions. There is no comparison available till date based on the morphology of the nanomaterial photocatalysts belonging to the same composition and with different similar nanomaterials. Hence an attempt was made by the authors of the present manuscript to highlight the effect of morphology of the photocatalyst on the reaction parameters. Based on the presently available literature the logical argument with in-depth analysis was presented which is backed with existing data from the different works. The available reviews on titanium dioxide, zinc oxide, silver phosphate, cadmium sulphide and zinc titanate photocatalysts put emphasis on the particular photocatalyst but do not show comparison within them or with other similar kind of photocatalysts. It is not only the morphology but entire spectrum of parameters of the photocatalysts were analysed and discussed thoroughly. The entire review reflects the holistic approach towards the comparison of the photocatalysts and morphology of the photocatalysts is one of the pointers which is reasonably discussed in the research article.

  1. The list of reference needs to be updated especially for those research findings since 2019 – 2022.

Ans: Thanks for the suggestion, the list of references is now updated by addition of latest references.

[3]       C. Zhao, M. Xi, J. Huo, C. He, Phys. Chem. Chem. Phys. 2021, 23, 23219.

[4]       M.R. Wang, L. Deng, G.C. Liu, L. Wen, J.G. Wang, K. Bin Huang, H.T. Tang, Y.M. Pan, Org. Lett. 2019, 21, 4929.

[5]       S. Nikazar, M. Barani, A. Rahdar, M. Zoghi, G. Kyzas, ChemistrySelect 2020, 5, 12590.

[190]   C. Yu, F. Chen, D. Zeng, Y. Xie, W. Zhou, Z. Liu, L. Wei, K. Yang, D. Li, Nanoscale 2019.

[192]   G.M. L., Metals (Basel). 2020, 10, 820.

[193]   S. Gautam, H. Agrawal, M. Thakur, A. Akbari, H. Sharda, R. Kaur, M. Amini, J. Environ. Chem. Eng. 2020, 8.

[196]   A. Yumashev, B. Åšlusarczyk, S. Kondrashev, A. Mikhaylov, Energies 2020, 13.

  1. The recent literature review on modified titanium dioxide, zinc oxide, silver phosphate, cadmium sulphide and zinc titanate with responsive band gap energy and their photocatalysis performance should be summarized in a table as benchmarking purpose and discussed in detail with your review summary findings.

Ans: The following table is added to the manuscript.

Table 7. Important characteristics of photocatalytic nanomaterials TiO2, ZnO, Ag3PO4, CdS and ZnTiO3.

S. No.

Photocatalytic materials

Band Gap Energy

(Eg) eV

Photocatalytic performances

Ref.

1.

TiO2

Eg Ì´ 3.1

The synthesized TiO2 nanotubes possess better photocatalytic activity than the as-prepared counterparts because of the larger surface area and good crystallinity. Emission stability of the catalyst also validates that TiO2 nanotubes could find potential application in the cold-cathode-based electronics. Literature shows that the decoration of TiO2 nanotubes by noble metal nanoparticles (such as Au, Ag, and Pt) also enhance its photocatalytic activity.

[44]

[197][198]

2.

ZnO

Eg Ì´ 3.35

Qu et al. Synthesized various shapes and morphology of ZnO nanomaterials using different ultrasonic processes. The study revealed that ZnO nanoflower morphology shows the excellent photocatalytic activity.

[199]

 [200]

3.

Ag3PO4

Eg Ì´ 2.45

Morales et al. synthesized silver phosphate microcrystals with polyhedral morphologies and its higher surface area played important role for higher photocatalytic activity.

Geng et al. successfully prepared polyhedral morphology of Ag3PO4 microcrystal structure. The photocatalytic activity study confirmed its excellent photocatalytic ability.

[201]

[202]

4.

CdS

Eg Ì´ 2.4

Ganesh et al. synthesized CdS nanoflowers. The nano flowers of the CdS materials showed the better photocatalytic activity in visible light. Wang et al. recently studied Ti3C2 MXene@CdS based nanoflowers composites heterostructures. It shows steady photoluminescence intensity and longer fluorescence lifetime.

   [75]

[203]

5.

ZnTiO3

Eg ̴ 2.9 (indirect)

3.59 (direct)

Dutta et al. reported the Ag doped ZnTiO3 nanorods. The photocatalytic results show improved photocatalytic activity. Chuaicham et al. recently studied ZnTiO3 mixed metal oxide. Higher photocatalytic activity was observed for phenol photodegradation.

[182]

[204]

4) In my opinion, technical significance and novelty of the work is still lacking especially on the material such as titanium dioxide, zinc oxide, silver phosphate, cadmium sulphide and zinc titanate.  Authors need to study and review more on this kind of reported work.

Ans: Thank you for your suggestion. The review shows different perspective on photocatalytic performance based on important catalytic parameters. The detailed analysis based on different synthesis approaches used during the synthesis of various morphologies of the same composition of titanium dioxide, zinc oxide, silver phosphate, cadmium sulphide and zinc titanate were studied in detail. The critical analysis of the different morphologies within the same composition and outside similar other materials morphologies were analysed. The typical, well studied and important photocatalytic materials were chosen as representatives to bring home the point. We are hopeful that this approach will create interest in the mind of the readers. The study can stimulate further curiosity amongst about the several other materials based on the grounds used in the present work. Artificial intelligence (AI) programme can be developed based on the data given in this study for the selection of the photocatalysts.  

5) Furthermore, the details studies on future direction and commercialization perspectives for this potential photocatalysis system with different morphologies of specific nanomaterials in photocatalysis related applications should be determined and reported.

Ans:

2.7 Future direction for the application of nanomaterial photocatalyst

Presently, nanotechnology manipulates and creates matter to fabricate materials in nanometre scale. Nanomaterials are used in interdisciplinary field and they have societal implications for wide range of scientific and engineering disciplines. The past few decades nanotechnology was applied in various sectors like; environment, materials science, energy, electronics, agriculture, healthcare, biotechnology, and information technology. Certain nanomaterials have excellent prospect to revolutionize wastewater treatment by photocatalysis and other techniques. Because of its remarkable characteristics such as effective adsorbents and photocatalytic property nanomaterial oxides are being used commercially for wastewater treatment extensively [221]–[223]. For technology related to clean environment, and health care like drug delivery and cancer therapy, nanotechnology had commercially provided scope for existing firms to upgrade their products and services. Several oxides, sulfides, tantalates, phosphates and niobates are used commercially for multiple cures [224][225]. Silver phosphate and cadmium sulfide is used commercially as a photocatalysts in novel photo reactors for better catalytic activity [226].

  1. The photocatalysis mechanism of different morphologies of specific nanomaterials should be described and discussed accordingly.

Ans: Following table is included in the manuscript.

Table 6. Mechanism of photocatalysis for TiO2, ZnO, Ag3PO4, CdS and ZnTiO3 photocatalysts nanomaterials photocatalysts.

S.No.

Materials

Photocatalysis mechanism

Ref.

1.

TiO2

[192]

2.

ZnO

[193]

3.

CdS

[194]

4.

Ag3PO4

[195]

5.

ZnTiO3

[196]

Round 2

Reviewer 1 Report

The paper can be accepted

Reviewer 3 Report

In overall, this manuscript was technically well revised. This revised manuscript meets the criteria of Molecules. Therefore, in my opinion, the revised manuscript can be accepted for publication.